# Non-invasive in vivo hyperspectral imaging of the retina for potential biomarker use in Alzheimer's disease

Xavier Hadoux [ID] et al.[#]

Studies of rodent models of Alzheimer's disease (AD) and of human tissues suggest that the retinal changes that occur in AD, including the accumulation of amyloid beta (Aβ), may serve as surrogate markers of brain Aβ levels. As Aβ has a wavelength-dependent effect on light scatter, we investigate the potential for in vivo retinal hyperspectral imaging to serve as a biomarker of brain Aβ. Significant differences in the retinal reflectance spectra are found between individuals with high Aβ burden on brain PET imaging and mild cognitive impairment ($n = 15$), and age-matched PET-negative controls ($n = 20$). Retinal imaging scores are correlated with brain Aβ loads. The findings are validated in an independent cohort, using a second hyperspectral camera. A similar spectral difference is found between control and 5xFAD transgenic mice that accumulate Aβ in the brain and retina. These findings indicate that retinal hyperspectral imaging may predict brain Aβ load.

[#]A full list of authors and their affiliations appears at the end of the paper.

The accumulation of Aβ in the brain is a hallmark feature of AD[1,2]. Advances in positron emission tomography (PET) imaging with tracers that are selective for Aβ have great utility in confirming a clinical diagnosis of the disease, however the application of this approach is limited by its cost and accessibility. As a consequence, PET imaging is not presently scalable for population screening of individuals at risk of AD. Given that Aβ accumulates in the brain over 15–20 years prior to the onset of manifest cognitive impairment[3], the lack of a convenient in vivo biomarker of Aβ stands as a lost opportunity for early intervention.

The retina is a developmental extension of the brain[4,5] and is the only part of the central nervous system that can be imaged non-invasively at sub-cellular resolutions[6–14]. Human post-mortem histopathological studies have shown accumulation of Aβ in the retinas of those with confirmed AD, principally in the inner-retinal layers[15–17]. Aβ is also known to occur in the sub-retinal deposits of people with age-related macular degeneration[18]. Studies of transgenic mouse models of AD have demonstrated the presence of retinal Aβ[19–21] and shown quantitative and temporal correlations between brain and retinal Aβ deposition[15–17]. Although some disagreement remains regarding the extent to which retinal Aβ accumulation occurs and the extent to which it phenocopies accumulation in the brain[22,23], emerging evidence highlights the potential for retinal imaging as a biomarker of brain Aβ status.

In the search for convenient retinal biomarkers of AD, the use of orally administered curcumin, a fluorescent compound with affinity for Aβ, in conjunction with retinal imaging has shown promise in preclinical and early phase human trials[17,24]. Other work has attempted to identify evidence of Aβ in brain and retinal tissue without the need for a label using hyperspectral (HS) imaging[15,25]. The approach is based on the acquisition of a series of images of an object across many contiguous wavelengths of light. HS images are thus stacks of images that combine spectral and spatial information into a single data cube. Each HS pixel corresponds to the reflectance spectrum of a locus across all the measured wavelengths (Fig. 1). Preclinical studies using HS microscopy of brain and retinal tissue from transgenic AD model mice have shown that Aβ exerts a characteristic influence on reflectance of light (spectral signature) and that the magnitude of this effect varies in proportion with the amount of Aβ[15]. Similar effects were found with HS microscopy of human brain and retinal tissue from individuals with confirmed AD, but not in those without the disease[15]. The application of this technology to image the eye in vivo has proven challenging, however, recent work describes a spectral signature of Aβ in AD-model mice using a dark-field HS endoscopy[25]. Here, we report the first clinical application of HS imaging in humans with and without moderate–high brain Aβ load.

## Results

**Participant recruitment and demographics.** To elucidate the spectral changes in Alzheimer's disease (AD) with retinal HS imaging, we recruited a cohort of participants who had undergone Aβ PET scanning and neuropsychological tests (principal cohort). Three participants (Aβ PET+) were excluded from the study owing to posterior capsular opacification after artificial lens implantation that prevented successful imaging. The recruited cohort consisted of 35 participants who were assigned to either the case group (Aβ PET+, $n = 15$) or the control group (Aβ PET−, $n = 20$), on the basis of brain Aβ PET scan results. All but one participant in the control group (owing to excessive blinking) had both eyes successfully imaged using the Metabolic Hyperspectral Retinal Camera (MHRC, Optina Diagnostics, Montreal, Canada).

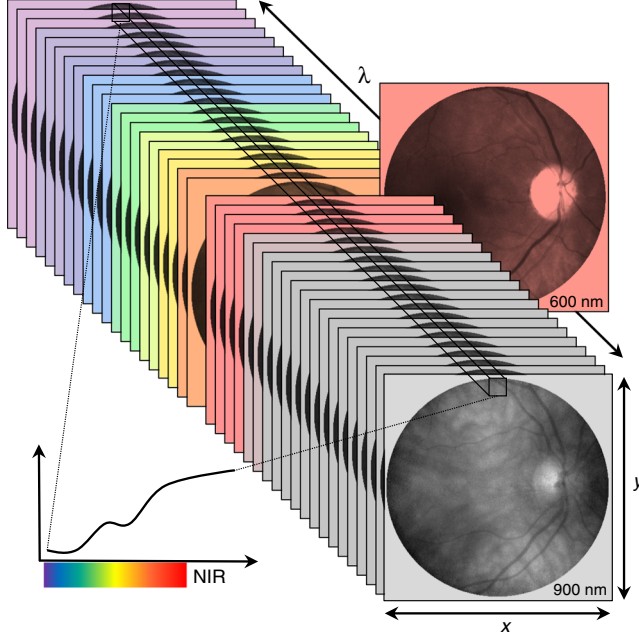

**Fig. 1** Principle of retinal hyperspectral (HS) imaging. In HS imaging a narrow bandwidth tunable light source illuminates the retina and the reflected light from the retina is collected by an image sensor. The different frames of the hyperspectral reflectance cube are obtained by scanning the source wavelengths. Therefore, each HS image has both spatial and spectral information, i.e., each spatial locus has an associated spectrum when viewed across the available wavelengths. NIR = near-infra-red

There were no differences between the groups for age and sex (Table 1). On average, controls scored higher in the mini mental state examination (MMSE, $p < 0.0001$, unpaired two-tailed $t$ test) compared with the cases (Table 1). No differences were found between groups for lens status, the presence of macular and peripheral drusen, presence of glaucoma, or retinal nerve fibre layer (RNFL) thickness on the basis of clinical examination, colour fundus photography and optical coherence tomography (OCT, Table 1).

**Uncorrected data do not show significant group differences.** To account for within-subject variability and avoid selection bias, we systematically sampled six regions of the retina based on well-defined anatomical landmarks (Fig. 2e). The raw reflectance spectra for each sampling location in the principal cohort are shown in Supplementary Fig. 1A–F. These spectra exhibit the characteristic pattern of fundus reflectance with low reflectance in the blue/yellow wavelength range (450–580 nm), increasing in the orange/red (590–760 nm) and flattening in the infra-red (770–900 nm)[26]. However, owing to the large dynamic range of fundus reflectance (approximately two orders of magnitude from 450–900 nm), raw reflectance spectra are not useful for visualisation of group differences (Supplementary Fig. 1A–F). Reflectance spectra centred about the average spectrum of all the participants in the principal cohort (Aβ PET+ and PET−) are displayed in Fig. 2f–k, highlighting the difference in spectra observed at the six sampling locations owing to variations in retinal structures at each location. As expected, a great degree of inter-subject spectral variability was also present. Although a trend was observed at every location, no statistically significant differences were found between cases and controls on the basis of uncorrected reflectance data (Supplementary Fig. 2). The difference between cases and controls observed at wavelengths close to 550 nm (Supplementary Fig. 2) approached statistical significance

**Table 1 Participant demographics**

|  | Cases (Aβ PET+) n = 15 | Controls (Aβ PET−) n = 20 | p value | Effect size | 95% CI |
|---|---|---|---|---|---|
| Age* | 68.5 ± 8.1 | 69.1 ± 2.7 | 0.77 | 0.57 | −3.35–4.48 |
| Sex (female/male)† | 13/2 | 13/7 | 0.15 | 0.29 | 0.05–1.64 |
| Lens (phakic/pseudophakic)† | 12/3 | 19/1 | 0.17 | 0.21 | 0.02–2.27 |
| Macular drusen (yes/no)† | 3/12 | 5/15 | 0.73 | 1.33 | 0.26–6.74 |
| Peripheral drusen (yes/no)† | 3/12 | 5/15 | 0.73 | 1.33 | 0.26–6.74 |
| Glaucoma (yes/no)† | 1/14 | 2/18 | 0.73 | 1.56 | 0.13–18.96 |
| RNFL thickness (μm)* | 97.5 ± 12.7 | 92.9 ± 12.7 | 0.29 | −4.68 | −13.50–4.13 |
| MMSE* | 23.2 ± 3.5 | 27.8 ± 1.6 | <0.0001 | 4.6 | 2.8–6.4 |

PET positron emission tomography, RNFL retinal nerve fibre layer, MMSE mini mental state examination
*Continuous variables are expressed as mean ± standard deviation and analysed with an unpaired two-tailed t test. The effect size and corresponding 95% CI are that of the difference between means.
†Dichotomous variables are expressed as number of participants and analysed with chi-square test. The effect size and corresponding 95% CI are those of the odds ratio

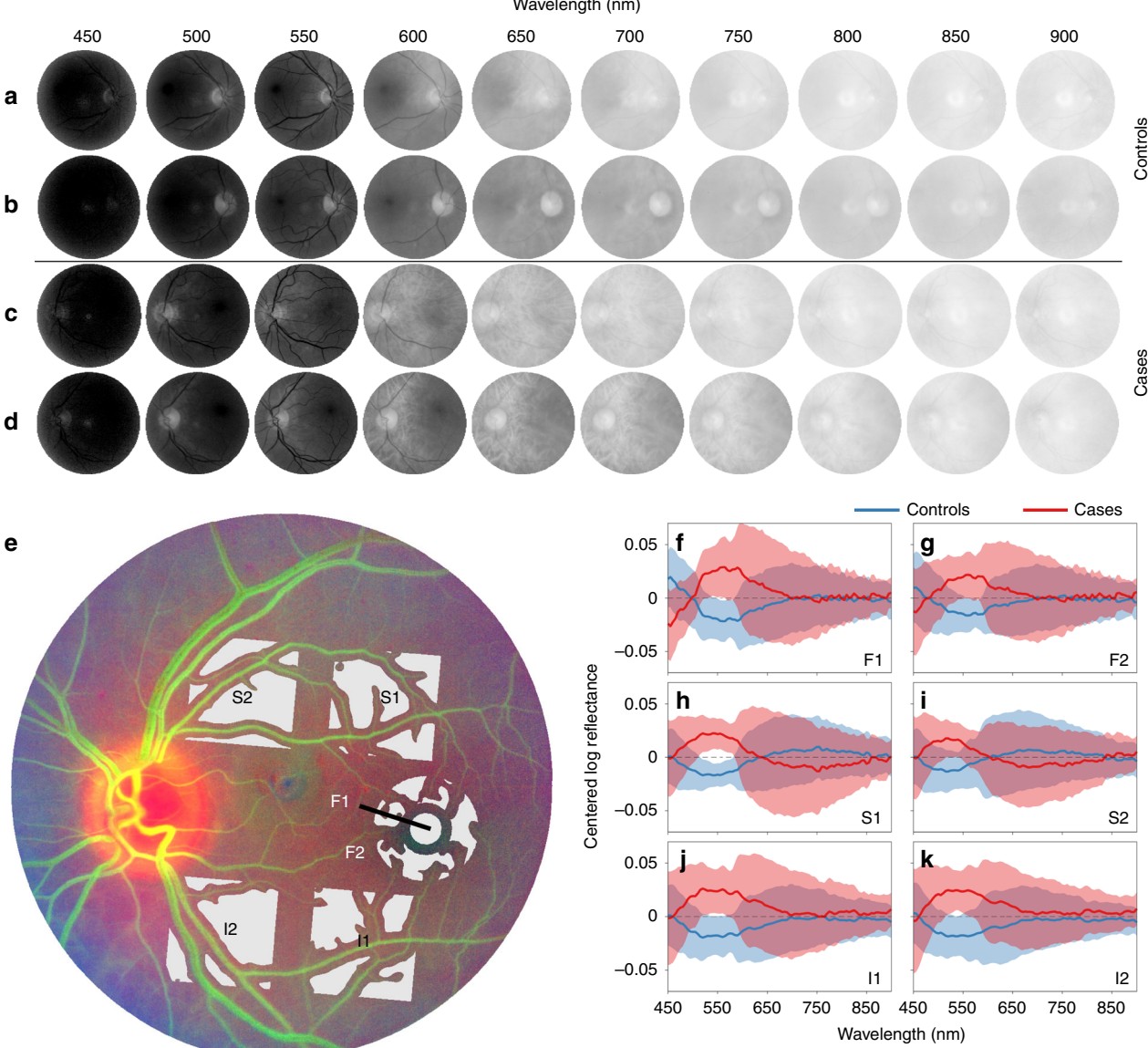

**Fig. 2** Spectral variation between eyes precludes discrimination between cases and controls. **a–d** Representative hyperspectral (HS) montages of four eyes (n = 2 controls, n = 2 cases from 450 to 900 nm in 50 nm steps) showing the inherent variability within- and between eyes owing to the retinal and choroidal vasculature, ocular pigment and ocular media. **e** Systematic sampling method used to analyse HS images including foveal locations (F1, F2) as well as locations superior (S1, S2) and inferior (I1, I2) to the temporal vascular arcades with segmentation of the major inner-retinal blood vessels. **f–k** Centred reflectance spectra at the different sampling locations for controls (n = 20, blue) and cases (n = 15, red) highlighting the large degree of inter-subject variability using uncorrected spectral data, which precludes discrimination between cases and controls. Centred spectra were obtained by subtracting the average spectrum from the spectrum measured for individual participants in the principal cohort (Aβ PET+ and PET−). Data shown as mean ± SEM

because the spectral variability within each group is lower in this wavelength range (Supplementary Fig. 3). These findings indicate that variability in key determinants of ocular reflectance must be accounted for before meaningful comparisons can be made between individuals for subtle spectral signatures, such as that reported for Aβ.

**Correction of within-group spectral variability**. To correct for sources of spectral variability unrelated to AD we used a spectral analysis method called Dimension Reduction by Orthogonal Projection for Discrimination (DROP-D; Supplementary Methods)[27]. The idea underpinning this technique is to identify the principal orthogonal axes of the data in parameter space (in this case, as a function of wavelength), which provides phenomenological measures of variations across the sample. This method was chosen as it is robust to over-fitting and is useful in situations where the sample size is small relative to the number of variables (number of wavelengths) studied. The DROP-D method identified the two main spectral components that account for most of the within-group variability and discrimination between cases and controls was optimal when these spectral components were corrected for (cross-validation, Supplementary Fig. 4). Regardless of the sampling location, these spectra (w1 and w2) were closely approximated with linear combinations of the known spectral influences of the ocular media (chiefly the lens)[28], macular pigment[29], melanin[26] and haemoglobin[30] (root mean square error across sampling locations, RMSE = 0.4 ± 0.1% (mean ± SD) for w1 and RMSE = 1.1 ± 0.4% for w2, Supplementary Figs. 5 and 6 and Supplementary Tables 1–6). This indicates that the corrected spectral components are largely determined by these ocular structures, substantiating the rationale for using them to correct the reflectance data. When this correction method is applied to HS images, variability owing to differences in fundus pigmentation and cataract are reduced (Supplementary Figs. 7 and 8). Reflectance spectra were different between cases and controls at shorter wavelengths (<565 nm) after correcting for these mundane sources of variability (Fig. 3b, c).

The main spectral difference between cases and controls was then computed with the DROP-D method as the principal axis of the between-group covariance matrix[27] (Fig. 3d). This constitutes the spectral model, which was used to summate reflectance measured at the 91 wavelengths used for illumination into a single HS score for each participant (inner product; Supplementary Methods 1). Although the HS score is a weighted summation of all the wavelengths according to the model intensity (Fig. 3d), the spectral information at which the groups are the most different (Fig. 3c), (i.e., shorter wavelengths <565 nm) will contribute most to the scores. For brevity, the derived model for only one sampling location (S1) is shown here. Models for the other sampling locations are shown in Supplementary Fig. 9.

**HS score discriminates Aβ PET+ cases from PET− controls**. The spectral model derived for each sampling location was used to calculate a HS score for each participant, for each location (Fig. 4). Overall, HS scores were higher for cases than for controls across retinal locations ($F_{1,33}$ = 7.1, $p$ = 0.01, two-way repeated measures analysis of variance), indicating a general spectral difference between cases and controls, which was only visible after correcting for the inherent spectral variation between eyes.

Using pairwise comparisons at each location, controlled for false discovery rate, we showed that the most significant difference between groups was found at the S1 sampling location ($p$ = 0.002, 95% CI: 0.06–0.22, unpaired two-tailed $t$ test, Fig. 4). Significant differences between cases and controls were also found at the F1 location ($p$ = 0.02, 95% CI: 0.04–0.36, unpaired

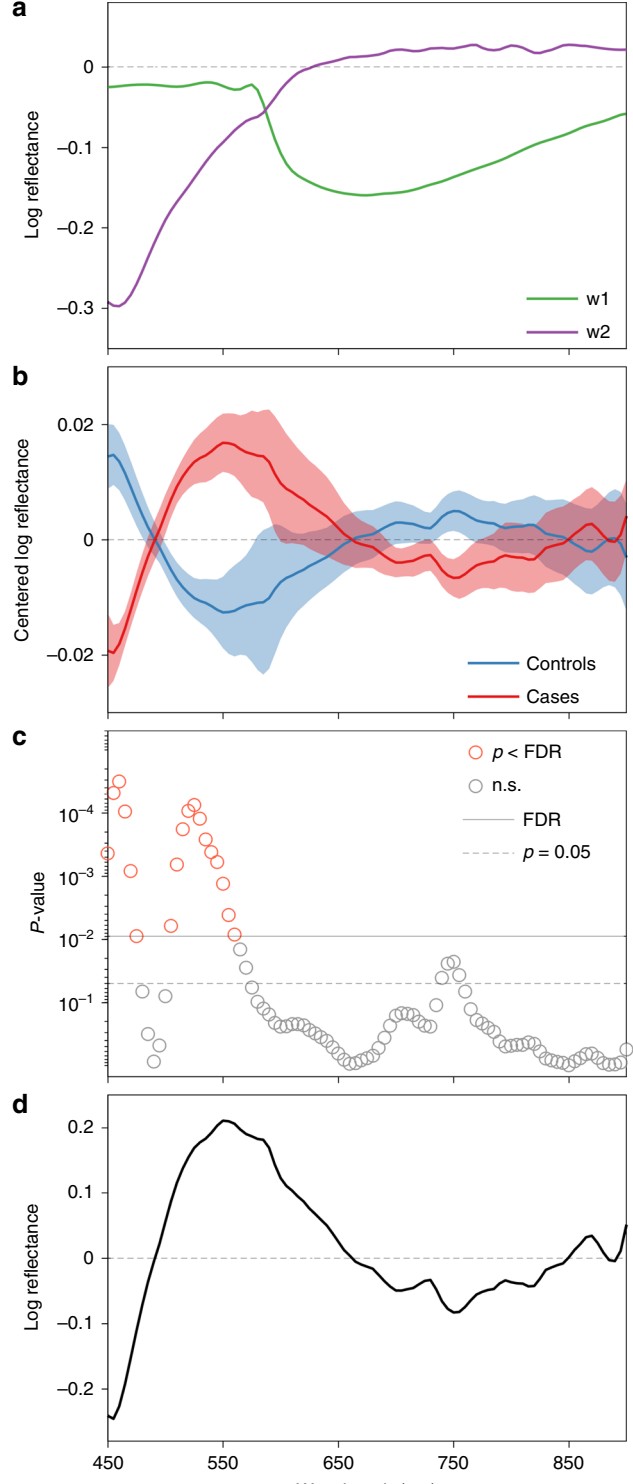

**Fig. 3** Identifying a spectral difference between groups. **a** Estimation of two main spectral components of within-subject variability (w1, green and w2, purple). These components were largely explained by a combination of spectra of known ocular constituents and were used to correct each reflectance spectrum. **b** Corrected reflectance spectra at sampling location S1 ($n$ = 20 controls, $n$ = 15 cases, data shown as mean ± SEM) revealed a spectral difference between the two groups. **c** P values for two-sided unpaired $t$ tests between groups using false discovery rate (FDR) control for significance across all the wavelengths (n.s. is for non-significant). **d** Spectral model at sampling location S1 corresponding to the main spectral difference between the two groups

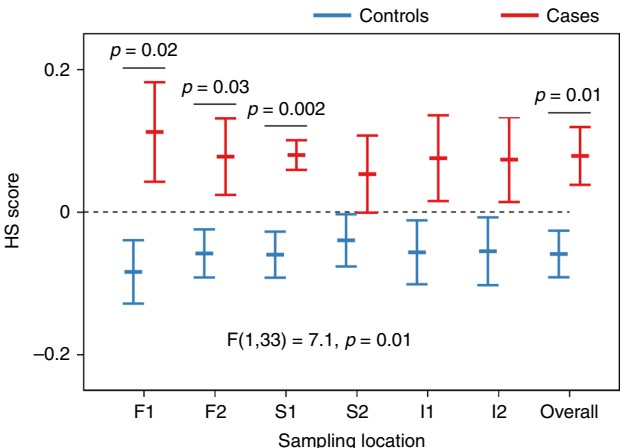

**Fig. 4** Discrimination between case and control groups using hyperspectral (HS) scores. HS scores obtained at the different sampling locations were higher for cases than for controls ($F_{1,33} = 7.1$, $p = 0.01$, two-way repeated measures ANOVA). Pairwise multiple comparisons (two-sided unpaired $t$ tests controlled for false discovery rate for each comparison) show significant differences between case ($n = 15$) and control ($n = 20$) groups at sampling locations F1, F2, S1 and for the average of all retinal locations (overall). Data shown as mean ± SEM. Source data are available as a Source Data file

two-tailed $t$ test, Fig. 4), F2 location ($p = 0.03$, 95% CI: 0.01–0.26, unpaired two-tailed $t$ test, Fig. 4) and for the average of all retinal locations ($p = 0.01$, 95% CI: 0.03–0.24, unpaired two-tailed $t$ test, Fig. 4). No significant correlations were found between retinal HS score and age ($r = 0.18$, $p = 0.3$, 95% CI: −0.17–0.48), MMSE ($r = −0.29$, $p = 0.1$, 95% CI: −0.57–0.05) or RNFL thickness ($r = 0.15$, $p = 0.4$, 95% CI: −0.2–0.48). HS scores were not significantly influenced by the presence of drusen in the case ($p = 0.30$, 95% CI: −0.15–0.05, unpaired two-tailed $t$ test) or control groups ($p = 0.51$, 95% CI: −0.19–0.10, unpaired two-tailed $t$ test).

**Method validation.** To validate the derived HS score as a potential biomarker for discrimination between Aβ PET+ cases and PET− controls, we applied the spectral model to additional validation data sets at the sampling location (S1) where the largest group difference was found. All subsequent references to HS score relate to the S1 location unless specified otherwise. Validation data sets included the fellow eye of each participant and an independent case (Aβ PET+) and control (Aβ PET−) cohort imaged on a separate MHRC using the same acquisition protocol (Fig. 5a).

Significant differences in HS scores were found between all case and control data sets. In the principal cohort, HS scores for fellow eyes of cases were significantly higher than the fellow eyes of controls ($p = 0.02$, 95% CI: 0.02–0.21, unpaired two-tailed $t$ test, Fig. 5a). Similarly, in the validation cohort HS scores of cases were significantly higher than those of controls ($p = 0.03$, 95% CI: 0.02–0.38, unpaired two-tailed $t$ test, Fig. 5a).

To examine the performance of HS scores in discriminating Aβ PET+ cases from PET− controls at the individual level, receiver operating characteristic curves (ROC) for the principal (Fig. 5b, black curve) and validation cohorts were calculated (Fig. 5b, orange curve). The area under the ROC curves (AUC) in both the principal (AUC = 0.82, $p = 0.001$, 95% CI: 0.67–0.97, Fig. 5b) and validation cohorts (AUC = 0.87, $p = 0.03$, 95% CI: 0.69–1.0, Fig. 5b) demonstrate the effectiveness of the HS score in discriminating between cases and controls.

Post-hoc analysis of the association between quantitative Aβ PET load in brain and the derived retinal HS score shows a moderate positive linear correlation between the two metrics ($r = 0.46$, 95% CI: 0.13–0.69, $p = 0.008$, Fig. 5c).

**HS scores in fellow eyes.** To test the robustness of the model, we measured the correlation between the HS scores of the study and fellow eyes of each individual in the principal cohort. In some individuals, the fellow eye had more ocular pathology (e.g., presence of drusen) than the study eye. Despite these differences there was a significant correlation ($r = 0.6$, $p = 0.0002$, 95% CI = 0.32–0.78) between the HS scores for study and fellow eyes (Fig. 6a), indicating that the HS score is robust to a degree of variation in ocular pathology.

**HS scores in eyes before and after cataract surgery.** In addition, we investigated the effect of lens status on the HS score as, if age-related lens changes influence the score, this might limit the clinical utility of this biomarker. We recruited individuals undergoing routine surgery for visually significant cataracts and performed imaging prior to and following cataract extraction and artificial intraocular lens implantation (Supplementary Table 7). No significant group differences were found in HS scores pre- and post-operatively ($p = 0.88$, 95% CI = −0.15–0.18, two-sided paired $t$ test). Importantly, a strong correlation was found between HS scores measured pre- and post-operatively ($r = 0.8$, $p = 0.01$, 95% CI: 0.23–0.94, Fig. 6b), indicating that lens status (cataract or artificial intraocular lens) had minimal effect on the derivation of HS scores using the DROP-D method.

**Intra-session repeatability of HS score is high.** Repeat retinal HS images in the same imaging session by the same operator were captured for 25 participants to assess intra-session repeatability. HS scores were computed for each location and the correlation between repeat images was calculated. HS scores were strongly correlated at all sampling locations (Supplementary Fig. 10).

**Retinal HS imaging in AD-model mice.** Nine to fourteen-month old 5xFAD mice ($n = 12$) and non-transgenic littermate controls ($n = 10$) underwent retinal imaging using a custom-made HS camera (450–680 nm). This was done to establish whether the retinal spectral difference between human subjects with high and low brain Aβ load is also seen between mice when the same data processing method is applied. Retinal images of one mouse (5xFAD) had spectral intensities below three median absolute deviations for >50% of the recorded wavelengths and were excluded from further analysis. Owing to the difference in the wavelength ranges of the human and mouse HS-imaging systems, recalibration of the human spectral model was needed to calculate HS scores acquired with the mouse imaging system. Recalibration did not remove the ability to distinguish between human Aβ PET+ and PET− subjects (Supplementary Fig. 11). When the recalibrated spectral model was applied to the mouse imaging data, a significant group difference was found between 5xFAD and control mice ($p = 0.003$, 95% CI: 0.026–0.11, unpaired two-tailed $t$ test, Fig. 7). 5xFAD mice had higher HS scores than control mice, in keeping with the human findings. The ROC curve indicates that the HS score can be used to distinguish 5xFAD from control mice at the individual level (AUC = 0.86, $p = 0.005$, 95% CI: 0.65–0.99, Fig. 7). Immunohistochemistry for human Aβ was performed on retinal sections from three 5xFAD and one control mouse that had undergone antemortem in vivo retinal HS imaging (Fig. 7). Sparse deposits of Aβ both in the inner and outer retina were found in all 5xFAD mice but in no

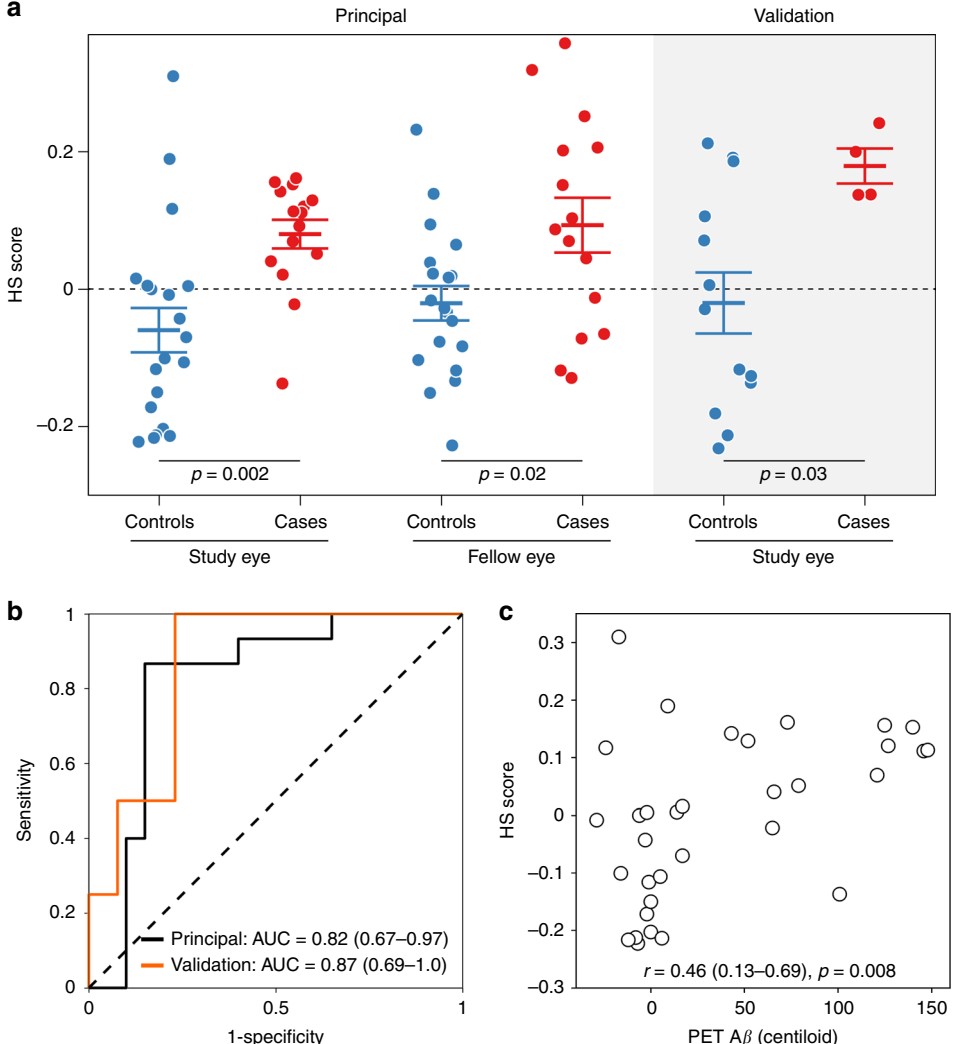

**Fig. 5** Validation of the spectral model. **a** Hyperspectral (HS) scores obtained for each data set (mean ± SEM). HS images of the principal cohort study eyes ($n = 15$ cases and $n = 20$ controls) and fellow eyes ($n = 15$ cases and $n = 19$ controls); and the validation cohort ($n = 4$ cases and $n = 13$ controls) were acquired on two different cameras of the same model, using the same imaging methods. Significant differences (two-sided unpaired *t* tests controlled for false discovery rate for each data set) were found between cases and controls in both cohorts. **b** Receiver operating characteristic curve (ROC) and area under the curve (AUC) for the principal cohort (black) and for the validation cohort (orange) show good discrimination between cases and controls. **c** Scatterplot of quantitative PET Aβ loads and HS scores (principal cohort study eye) showing significant positive correlation between the two metrics ($n = 33$). Source data are available as a Source Data file

sections from the control mouse. The corresponding HS scores for these mice are shown in Fig. 7.

## Discussion

This is the first in vivo human clinical study to report a label-free, non-invasive biomarker that can discriminate between Aβ PET+ case and PET− control participants. Owing to considerable inherent variability in the major determinants of ocular reflectance between individuals, and between retinal locations within individuals, raw retinal reflectance spectra were not useful for discrimination between cases and controls. The main axes of within-group spectral variability were removed and significant differences between the reflectance spectra of cases and controls were observed at various wavelengths, with the greatest differences apparent at shorter wavelengths (<565 nm). Spectral representations of the principal axes of variation that were adjusted for in this correction were closely approximated with linear combinations of the known spectra of the ocular media

(chiefly the lens), macular pigment, melanin and haemoglobin. Following correction, we used a simulation to show that much of the difference between cases and controls could be described by the spectral profile of Aβ in solution (Supplementary Methods; Supplementary Discussion; Supplementary Figs. 12 and 13, Supplementary Tables 8 and 9). Weighted differences across wavelengths were computed for each participant and summarised as a HS score. HS scores were significantly different between cases and controls at the group level (validation cohort, $p = 0.03$, 95% CI: 0.02–0.38, unpaired two-tailed *t* test, Fig. 5a). High-test performance was demonstrated for classification of individual participants (validation cohort, AUC = 0.87, 95% CI: 0.69–1.00, Fig. 5b). Although the DROP-D method is optimised for discrimination as opposed to quantification[27], HS scores were nonetheless associated with brain Aβ burden as measured with PET imaging ($r = 0.46$, 95% CI: 0.13–0.69, $p = 0.008$, Fig. 5c). Although the latter finding is interesting, further studies are needed to establish the extent to which retinal HS scores are quantitatively associated with brain Aβ levels. HS scores were

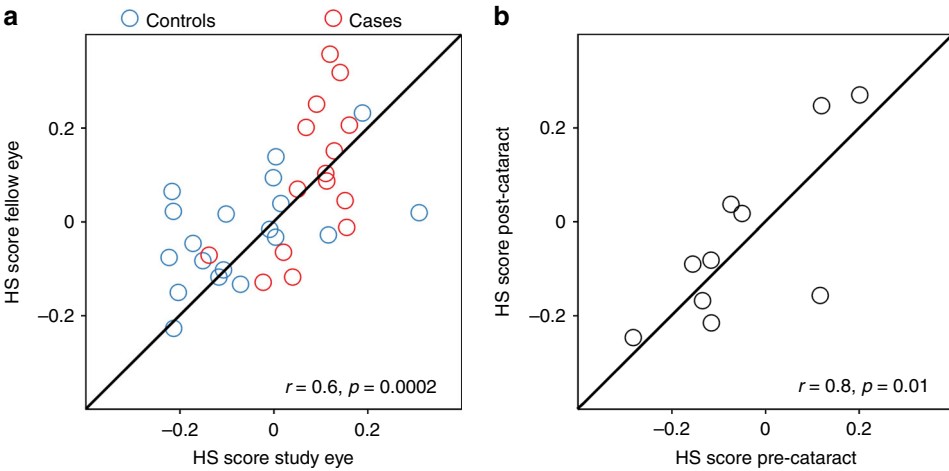

**Fig. 6** Hyperspectral (HS) scores show robust agreement between eyes and before and after cataract surgery. **a** Correlation showing good agreement between the HS scores of the study and fellow eye ($n = 15$ cases, red; $n = 19$ controls, blue). **b** Correlation showing good agreement between the HS scores pre- and post-cataract surgery ($n = 10$ independent participants). Source data are available as a Source Data file

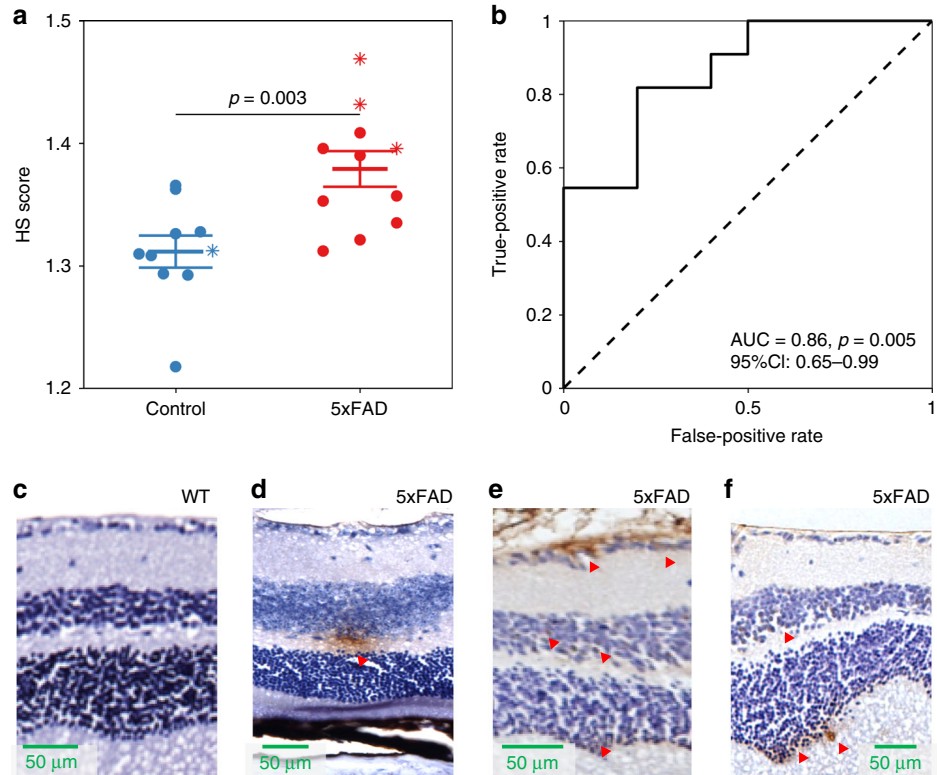

**Fig. 7** 5xFAD retinal HS imaging validates human findings. Mouse retinal hyperspectral (HS) scores were calculated using the recalibrated human spectral model (450–680 nm). **a** HS scores were significantly different for control (blue, $n = 10$) and 5xFAD mice (red, $n = 11$; mean ± SEM). The markers (*) denote HS scores of mice that were selected for retinal Aβ immunohistochemistry **c–f**. **b** Receiver operating characteristic curve (ROC) and area under the curve (AUC) showed good discrimination between control and 5xFAD mice. **c–f** Immunohistochemistry of 5xFAD retina using human Aβ-specific 1E8 antibodies. **c** shows a wild-type (WT) mouse. **d–f** show representative 5xFAD mouse retinas (age 9–14 months). Brown regions denoted by red arrows mark regions of Aβ immunoreactivity. Source data are available as a Source Data file

highest in the superior (S1) and foveal (F1 and F2) sampling locations (Fig. 4). Little is known of regional variations in the distribution of Aβ in the human retina. One human study has identified increased Aβ plaque distribution in the superior retina, however, the generalisability of this finding and its pathophysiological basis are not known[17]. The application of our human spectral image processing method to retinal images of mice enabled reliable discrimination of transgenic AD-model mice

from matched controls, when calibration was applied to account for the narrower wavelength range of the mouse imaging system. The major difference between these matched inbred mouse strains is overexpression of Aβ in 5xFAD mice and Aβ is known to accumulate in the retinas of these mice (Fig. 7)[31]. Accordingly, it is likely that the HS imaging signature identified in both AD-model mice and in human subjects with high brain Aβ loads is attributable to retinal Aβ.

We do not have conclusive evidence that the HS score is owing to retinal Aβ alone. It is possible that the spectral effect observed in this study was due to other factors that are associated with brain PET Aβ status, such as iron accumulation, tau phosphorylation or inflammation[31–33]. Any ocular constituent with a similar spectral profile to that measured for Aβ in solution could account for the observed effect (Supplementary Fig. 12), provided that it was differentially distributed between Aβ PET+ cases and Aβ PET− controls. For instance, a recent large longitudinal study found that subtle changes in thickness of the retinal nerve fibre layer, measured using OCT, were associated with cognitive decline[34]. It is possible that these and related structural alterations in the retina in AD may contribute to the detected spectral signature, however no such changes were detected using OCT in the present study. Although OCT studies have identified associations between retinal nerve fibre layer thinning and AD, these effects are small in magnitude, non-specific and not predictive at the individual level[35]. Another limitation of the work is the small number of subjects in the validation cohort. Nevertheless, our method could, in the absence of alteration, accurately predict brain Aβ PET status in the validation cohort and in a cohort of AD-model mice.

Given the high prevalence of cataract and cataract surgery in the population at risk of AD, we tested whether lens status adversely affected test performance. Strong correlation between pre-operative and post-operative HS scores indicated that the test was robust to visually significant cataracts and to artificial intraocular lens implantation. This finding is important as it is a precondition to utility of the test in clinical practice.

This study examined a cohort of people that were similar for demographic and ophthalmic characteristics, for whom Aβ PET status was known. Bias was reduced through the use of a semi-automated method for sampling retinal loci. The spectral model that was developed to discriminate between cases and controls could be explained by known influences on ocular reflectance, as well as Aβ, and thus has biophysical plausibility. Although the subject numbers were small, the difference between groups was significant and this was confirmed in an independent cohort imaged on a separate camera. Additional work is underway to examine the quantitative and topographic relationships between the retinal HS score and brain Aβ burden.

Koronyo et al.[17] describe the use of an exogenous fluorophore with binding affinity for Aβ in combination with fluorescence retinal imaging in vivo in humans. Although this approach has potential as a biomarker of AD, the existing evidence is drawn from a cohort with clinically diagnosed AD without PET imaging. Further studies are awaited. HS imaging has potential advantages over this approach as it can be administered at the point-of-care without the need for contrast agents. The HS method is rapid, well tolerated and can be repeated at short intervals, as necessary. Near real-time semi-automated analysis of the HS data cube is possible at the time of image acquisition.

Further work is needed to establish the utility of this imaging approach in people with eye diseases such as glaucoma[36] and age-related macular degeneration[37]. Although the method has been demonstrated to be robust to moderate visually significant cataract, dense cataracts that preclude retinal imaging will not be compatible with this approach. Although some disagreement remains regarding the extent to which retinal Aβ accumulation occurs in AD, an increasing number of studies support this assertion[15–17]. These studies use immunohistochemistry on post-mortem human tissues with validated antibodies directed at different epitopes of Aβ, complemented by transmission electron microscopy and show characteristic features of Aβ accumulation. Additional studies, using standardised methods, are required to demonstrate the extent to which retinal changes parallel brain changes in AD.

In summary, we have identified a non-invasive retinal imaging method to discriminate between individuals with and without moderate–high brain Aβ burden using a commercial camera together with data analysis methods. This approach may eventually have value for screening individuals at risk AD and may have a role in monitoring disease progression.

## Methods

**Study design.** The primary objective of this study was to determine whether retinal HS imaging could be used to distinguish Aβ PET+ (case) from PET− (control) participants. Accordingly, participants who had undergone brain Aβ PET imaging were recruited for HS imaging of the retina. A spectral analysis method was used to minimise spectral variability and identify a difference between Aβ PET+ cases and PET− controls. The method was further validated against an independent cohort to determine whether the same discrimination performance could be obtained. To understand the underlying components of the main spectral difference between groups, modelling was performed using a combination of spectra of ocular structures that are known to affect ocular reflectance as well as the spectrum of Aβ in solution derived from spectrophotometry. This information was used to ascertain whether an Aβ signature could be seen with HS imaging of the retina in Aβ PET+ participants.

**Ethics.** All protocols and methods described in this study were approved by the Human Research Ethics Committee of the Royal Victorian Eye and Ear Hospital (Project number 15/1253 H and 16/1265 H) and by Veritas IRB for the validation cohort in Canada (16093) and were conducted in accordance with the Declaration of Helsinki. Informed consent was obtained from all participants following a thorough explanation of all test procedures. All procedures involving mice were performed in accordance with the ARVO Statement for the Use of Animals in Ophthalmic and Vision Research. Ethics approval was obtained from the Howard Florey Institute Animal Ethics Committee (Approval number: 13-068-UM).

**Participant recruitment.** Participants in the principal cohort were recruited from two sources, the Royal Melbourne Hospital Neuropsychiatry Unit and the Australian Imaging, Biomarker & Lifestyle Study of Ageing (AIBL) study. All cases had either mild cognitive impairment or early AD as determined by a neuropsychological test battery and moderate–high Aβ burden measured on brain PET imaging prior to study commencement. The tracer-specific standardised uptake value ratio (SUVR) diagnostic thresholds for PET positivity were PiB ≥ 1.40, NAV4694 ≥ 1.40, Flutemetamol ≥ 0.55 and Florbetapir ≥ 1.05. The control group included age- and sex-matched participants with Aβ burden on PET scan below the diagnostic threshold for AD, or where Aβ distribution was atypical for AD.

Exclusion criteria included age under 30 years; opacification of the ocular media such as visually significant cataract (associated with visual acuity 6/18 or worse) or corneal disease sufficient to preclude retinal imaging; angle-closure glaucoma or angle-closure suspects; retinal surgery in the preceding 6 months; previous major ocular trauma; retinal dystrophy and participants with any medical conditions likely to preclude satisfactory retinal imaging.

**Imaging procedures.** All participants underwent a comprehensive ocular examination, by a registered optometrist (FH) or ophthalmologist (PvW) prior to imaging to identify any ocular exclusion criteria. Conventional retinal imaging techniques and clinical examinations were employed to assess the presence of other retinal pathologies and to evaluate comparability between groups. This is important as retinal nerve fibre layer thinning may exert confounding spectral influences[36] and as the subretinal deposits (drusen) that are found in age-related macular degeneration have been reported to contain Aβ, among other spectrally active constituents[37]. Different retinal sampling locations were used to assess the effect of these potential confounders. Optical coherence tomography (OCT) images were collected on a Spectralis SD-OCT (Heidelberg Engineering, Heidelberg, Germany) using three standard protocols: a horizontal posterior pole raster scan; peripapillary retinal nerve fibre layer scan; and optic nerve head (ONH) scan with Enhanced Depth Imaging. Colour retinal images were collected using the Digital Retinography System (Centervue SpA, Italy). To ensure optimal quality retinal images, pupils were dilated (>5 mm) using tropicamide 0.5% w/v (Bausch & Lomb, iNova Pharmaceuticals, NSW, Australia) and phenylephrine 2.5% w/v (Bausch & Lomb, iNova Pharmaceuticals, NSW, Australia).

HS imaging was performed using the Metabolic Hyperspectral Retinal Camera (MHRC, Optina Diagnostics, Montreal, Canada). The MHRC captures images of the retina with a 30-degree field of view and a pixel resolution of ~8.3 μm. Ninety-one consecutive frames were recorded for each HS image, scanning from 900 nm to 450 nm in steps of 5 nm. Exposure time was set to 0.01 s per frame, leading to an overall acquisition time of under 1 s (Supplementary Movie 1). The rapid acquisition was well tolerated by participants. Repeat imaging was performed when eye movement or blinks compromised image acquisition. The focal plane used to commence imaging was the inner-retinal blood vessels at an illuminating wavelength of 905 nm. Each image was vertically centred on the optic nerve and included the fovea.

Reproducible spectral imaging was ensured by measuring the power at the exit of the wavelength selection filter during each acquisition. Reflectance HS images were obtained by correcting the raw data for the dark current and internal reflection[38]. Light intensity at each illuminating wavelength was normalised using a model eye lined with Spectralon as per the manufacturer's instructions. The residual motion between adjacent frames was then registered to ensure that each pixel corresponded to the same spatial location on the retina for all wavelengths.

**Systematic sampling.** Sampling locations were selected semi-automatically for reproducibility and to avoid selection bias, as described below. A user (XH) was presented with a high contrast false-coloured image of the retina (Fig. 2e). The increased colour contrast representation of the fundus posterior pole was obtained by merging the spectral bands, which enhanced the visibility of the vasculature, fovea and ONH. The user manually selected the centre of the fovea and ONH and the orientation of the temporal raphe of the retinal nerve fibre layer (line between ONH centre and foveola) was calculated. Six sampling locations corresponding to different retinal anatomical structures were automatically extracted for each participant, after adjusting for the temporal raphe.

The foveola was sampled using a circular area of interest (60-pixel diameter) in the centre of the fovea (F1). The parafoveola was sampled using an annulus (100–200 pixels in diameter) centred on the fovea (F2). Areas in the temporal vascular arcades superior (S1) and inferior to the fovea (I1) as well as superior (S2) and inferior to the ONH (I2) were sampled using squares (200 × 200 pixels), on a fixed template horizontally orientated with the temporal raphe (Fig. 2e).

Given that retinal vascular anatomy varies considerably between individuals, each sampling location may contain varying numbers of pixels overlying blood vessels. Retinal blood vessels have a prominent spectral signature and thus failure to exclude vessels results in unnecessary variability in the spectral profile. In each sampling location, blood vessels visible in the inner retina were automatically segmented using the green channel of the false-coloured image. A difference of Gaussian (DoG)[39] image was calculated using variance parameters 1 and 20 to increase contrast between the blood vessels and background whilst removing noise and artefacts. To be conservative, 40% of pixels in the DoG image with the highest intensity were considered to be blood vessels[40].

**Spectral analysis.** The study eye, defined as the eye with the least amount of ocular pathology as determined by a trained clinician (FH, PvW) was used to derive the spectral model. The right eye was chosen in cases in which both eyes were similar. The PET status (positive or negative) of each participant in the principal cohort was used to develop the spectral model to discriminate between cases and controls. As the interaction between light and retinal tissue is highly dependent on the sampling location[26], each location was analysed independently. To transform the multiplicative interaction between retinal structure and reflectance, measured spectra were first log-transformed[41]. The average spectrum at each sampling location was then computed and smoothed using a Savitzky-Golay spectral smoothing procedure with a 5th order kernel of width of 13[42]. At each location, a machine learning method called Dimension Reduction by Orthogonal Projection for Discrimination (DROP-D)[27] was trained to classify Aβ PET− controls from PET+ cases on the basis of the spectral data from the study eye of the principal cohort. This linear machine learning discrimination method was used to derive the HS scores as it is robust to over-fitting and well-suited to problems in which the number of observations (participants) is small compared to the number of variables (wavelengths). In brief, DROP-D works in two steps (see Supplementary Methods for details). First, the method estimates and removes the main spectral axes of within-class variability from the data, i.e., spectral sources of variability that are detrimental to the discrimination problem. Second, it derives a model by computing the principal spectral axis of between-class variability on the corrected data. A HS score (single number) for a given spectrum is obtained by taking its inner product with the spectral model. Note that by construction, the HS scores are not sensitive to any linear combination of the removed within-class axis. To further avoid over-fitting, leave-one-out cross-validation was used to identify the number of spectral axes to optimise discrimination between groups. In addition, we modelled the shape of the removed spectral axes using linear combinations of the known spectrally active constituents of the eye including melanin[43], haemoglobin[44], ocular media[28] and macular pigment[29] (Supplementary Fig. 5). To test classification performance, the DROP-D model trained on the principal cohort data was also applied to fellow eye data; to image data of a validation cohort and to retinal image data of AD-model mice and controls.

**Biomarker validation.** The spectral model derived using the study eyes was applied to fellow eyes to provide internal validation of the model and to an independent cohort based in Montreal, Canada. One eye of each of 13 PET− controls and 4 PET+ cases of the independent cohort were imaged using a second MHRC of the same model (Optina Diagnostics, Montreal, Canada) and following the same imaging protocol. The tracer-specific SUVR diagnostic thresholds for PET positivity were Florbetaben ≥ 1.45. Between-group differences were tested using unpaired two-sided t tests, adjusted for false discovery rate for multiple comparison across all three data sets. The agreement of the HS scores between study and fellow eyes was estimated using Pearson's correlation coefficient. In developing the model

using data from the principal cohort, knowledge of Aβ PET status (positive vs negative) alone was known to the investigators. The quantitative PET data were provided (by co-author CJF) after the analytical model was finalised and correlation between HS score and PET quantitative values were performed in a blinded fashion. Blinding was not necessary for cross-validation studies (fellow eye study and validation cohort) as parameters of the analytical model were fixed.

**Biomarker robustness to lens status.** To assess the robustness of the model to the presence of lens opacification (cataract) and artificial intraocular lenses, a cohort of 10 participants scheduled to undergo elective outpatient cataract surgery were recruited from the Royal Victorian Eye and Ear Hospital, Melbourne, Australia. The Aβ PET status of these subjects was unknown and all were presumed to be cognitively normal. Retinal HS images were taken before and 8 weeks after cataract surgery. The spectral model obtained from the principal cohort was used to derive the HS scores pre- and post-cataract surgery. Differences in HS scores pre- and post-operatively were tested using a paired two-sided t test. The agreement of the HS scores pre- and post-operatively was estimated using Pearson's correlation coefficient.

**Mouse model and animal husbandry.** The 5xFAD mice[45] colony was established in-house using breeder pairs of 5xFAD mice (MMRRC Stock 34848, Jackson Laboratories, Bar Harbor, ME, USA) and C57BL/6 J mice. This strain (B6.CgTg (APPSwFlLon,PSEN1*M146L*L286V)6799Vas/Mmjax) backcrossed on a C57BL/6 J background does not carry the retinal degeneration allele Pde6b^rd1. Mice were genotyped using tail tissue samples shortly after birth by an external vendor (TransnetYX, Cordova, TN, USA). Non-transgenic littermates served as controls.

Mice were housed in temperature-controlled cages (20 °C) with ad libitum access to food and water and a 12-hour light–dark cycle. During the light cycle, ambient light was kept below 50 lux to avoid the possibility of photoreceptor damage (43). Twenty-two animals (5xFAD n = 12, WT n = 10) aged between 9 and 14 months old were imaged with the HS imaging camera. A subset of these (5xFAD n = 3, WT n = 1) were killed and retinal immunohistochemistry was performed.

5xFAD mice carry three human APP mutations and two PSEN mutations driven by the neuron-specific Thy1 promoter. These mice recapitulate aspects of AD pathology including the accumulation of high levels of Aβ in the brain and retina[31] as well as behavioural changes such as memory impairment[46].

On the B6SJL background (MMRRC Stock 34840-JAX), these mice are known to accumulate Aβ as early as 2 months of age in the brain and this increases with age[45,47]. 5xFAD mice on the C57BL/6 J background, as used in this study, are known to show less marked Aβ expression and develop Aβ plaques later in life (4–6 months). The advantage of this strain is the lack of an inherited retinal degeneration, which may affect the HS signal from the retina[45].

**Mouse retinal immunohistochemistry.** Immunohistochemistry was performed on retinas from three 5xFAD mice and one WT mouse, as previously described[48]. In brief, whole eyes were fixed in 10% neutral-buffered saline overnight and embedded in paraffin for sectioning. Sagittal sections (7 μm) were treated with 80% formic acid (5 min, room temperature) and 3% hydrogen peroxide (5 min, room temperature) prior to incubation (15 min, room temperature) in blocking buffer (50 mM Tris-HCl, 175 mM NaCl, pH 7.4, with 20% goat serum). Monoclonal mouse antibody 1E8 raised to human Aβ_{17−24} was used at a dilution ratio of 1:500 to detect total Aβ[49,50]. After washing, sections were incubated for 15 min at room temperature with biotinylated secondary antibody and streptavidin/horseradish peroxidase reagent (Dako LSAB + HRP kit, Agilent Technologies Australia, Mulgrave VIC, Australia, catalogue number K060911) prior to chromogen development (Dako DAB + chromogen kit, Agilent Technologies Australia, Mulgrave VIC, Australia). Slides were counterstained with Harris's haematoxylin (Australian Biostain, Traralgon, VIC, Australia), and mounted in distyrene-plasticiser-xylene media (DPX new, Merck Millipore, Bayswater, VIC, Australia). Sections stained with the omission of the primary antibody served as negative experimental controls.

**Mouse HS imaging.** Prior to imaging, animals were anaesthetised with an intra-peritoneal injection of 1 ml/100 g of a mixture containing 80 mg kg⁻¹ of ketamine (Ilium Ketamil 100 mg ml⁻¹, Troy Laboratory Pty Ltd, Smithfield NSW, Australia) and 10 mg kg⁻¹ of xylazine (Ilium Xylazil 100 mg ml⁻¹, Troy Laboratory Pty Ltd) and diluted in sterile injectable saline at a ratio of 1:10. Topical anaesthetic and mydriatic agents were instilled (Alcaine 0.5%, Mydriacyl 0.5% respectively, Alcon Laboratories, Frenchs Forest, NSW, Australia). Ocular gel (Genteal, hypromellose 0.3%/carbomer 980 0.22%, Novartis Pharmaceuticals Australia Pty Ltd, Macquarie Park, NSW, Australia) and a coverslip was placed over the surface of each eye to prevent corneal dehydration and to prevent cataract formation. A heated platform was used to maintain body temperature during imaging.

Mice were imaged using a custom-built bench ophthalmoscope comprised of a 150 W xenon light source and a fast switching monochromator (Polychrome V, Till Photonics, Hillsboro, OR, USA) oriented perpendicular to the pupil plane. A semi-reflective pellicle beam splitter (BP245B1, ThorLabs, Newton, NJ, USA) was used to combine the illumination and imaging paths. A condensing lens focused

the image onto a 16-bit monochromatic CMOS sensor (Andor Technology, Belfast, UK). Image acquisition was performed from 320 to 680 nm (1 nm step, bandwidth 10 nm). Images at each wavelength were acquired using a 100 ms exposure for a total acquisition time of 36.1 s. Images were exported in TIFF format.

**Mouse HS image processing.** Light intensity at each wavelength was normalised using a model eye lined with Spectralon. The residual motion between adjacent frames was then registered. Blood vessels visible in the inner retina were automatically segmented using an average image from wavelengths 390–460 nm. The mask used to extract blood vessels was derived using series of DoG[39] calculated using variance parameters 1–5 (0.2 step) for the small and 4–14 (1 step) for the large filters. This method increased contrast between blood vessels and the background while removing noise and artefacts. Pixels overlying blood vessels and the optic nerve were excluded. Owing to the smaller size of the mouse retina compared to human, the pixel intensities were averaged at each wavelength to yield a single reflectance spectrum per retina. The reflectance spectra were resampled between 450 and 680 nm in 5 nm steps using a linear interpolation method. The human HS model was recalibrated (450–680 nm) to suit the mouse imaging system as described above. Differences in illumination over the two imaging sessions were compensated for using the average spectral difference of control mice for each session. Retinal HS scores were obtained by applying the recalibrated human HS model to mouse reflectance spectral data.

**Statistics.** All analyses were performed using Matlab (R2017a, The MathWorks Inc., Natick, MA, USA) or Prism (v6.0, GraphPad Software Inc., La Jolla, CA, USA). Correlation was assessed using Pearson's correlation coefficient. A significance level of 0.05 was considered statistically significant. When multiple comparisons were performed, the family-wise significance level was maintained at 0.05 by adjusting the false discovery rate using the Benjamini–Yekutieli procedure[51]. The statistical tests used in this study are indicated in the results section and within figure and table legends.

**Reporting summary.** Further information on research design is available in the Nature Research Reporting Summary linked to this article.

## Data availability
Principal cohort clinical data were collected at the Centre for Eye Research Australia in collaboration with the AIBL study. Deidentified clinical data for the validation set were obtained from Optina Diagnostics. These data are not publicly available, and restrictions apply to their use. Source data underlying Figs. 4, 5, 6 and 7 are available as a Source Data file.

## Code availability
Described in Methods/Systematic sampling; Methods/Spectral analysis; Methods/Mouse Hyperspectral Imaging and in Supplementary Methods 1.

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

## Acknowledgements

We thank Mr Baillieu Myer and Mrs Samantha Baillieu for supporting this work. We also thank Dr Qiao-xin Li and Dr Holly Chinnery for their help with immunohistochemistry of mouse retinal sections. We appreciate the administrative assistance provided by Cheryl Donohue, Vasantha Pather Lowen and Vikki Marshall. We are grateful to investigators and staff affiliated with the Australian Imaging, Biomarker and Lifestyle Study of Ageing for assistance with study participant recruitment and for the provision of PET scan and clinical data. We are grateful for the generous support received from the Yulgilbar Alzheimer's Research Program, Pratt Foundation, Joan Margaret Ponting Charitable Trust, Hecht Charitable Trust (Perpetual), Coopers Brewery Foundation, Eldon & Anne Foote Donor Advised Fund and The Sylvia and Charles Viertel Charitable Foundation (VTL2015CO18; PvW was the recipient of a Clinical Investigatorship). PvW was the recipient of a University of Melbourne Annemarie Mankiewicz-Zelkin Fellowship. C.F. receives support from the Australian Research Council Centre of Excellence for Gravitational Wave Discovery (OzGrav; project number CE170100004). CERA receives Operational Infrastructure Support from the Victorian Government. The CQDM/Brain Canada Focus on Brain grant and the BLN-CE program, provided support for the Montreal cohort.

## Author contributions

P.vW., X.H. and F.H. designed and performed the experiments, analysed the data, wrote and edited the manuscript. J.K.H.L., B.V.B and C.T.O.N performed the mouse retinal imaging. A.P. and S.C. performed experiments examining Aβ in solution. J.H. collected data for the cataract study. S.L. assisted with participant recruitment and the analysis of neuropsychological data. C.M., C.J.F, C.R., V.L.V. assisted with recruitment of participants from the AIBL cohort; provided input into study design and analysed PET data. E.N.T., C. F. aided in data analysis and interpretation. J-P. Sylvestre, F.L., P.R-N., S.M. provided HS and PET data for the validation cohort. M.D. provided input into experimental design and assisted with funding applications. P.vW., X.H, J.G.C. and R.W. conceived the project, secured funding and advised on all studies. All authors edited the manuscript and approved the final version for publication. Validation cohort: J-P. Sylvestre, F.L., J-P. Soucy., P.R-N., S.G., J.D.A, M-A.R., S.B. designed the study; J-P. Sylvestre, J-P. Soucy, coordinated the cohort; P.R-N., S.M. performed the amyloid PET analysis; S.G., Z.N. recruited the participants and performed the neurological evaluation; J.D.A. and M-A.R. performed ophthalmological evaluation and MHRC imaging.

## Additional information

**Competing interests:** X.H., F.H., J.G.C. and P.vW. have filed an International (PCT) Patent Application No PCT/AU2019/000003 and are co-founders of Enlighten Imaging PTY LTD, a start-up company focused on developing novel retinal imaging solutions for neurological and retinal diseases. J-P. is employee at Optina and holds equity in Optina. J.D.A holds equity in Optina. The other authors declare no competing interests.

Xavier Hadoux [1,2], Flora Hui [1,2], Jeremiah K.H. Lim [3], Colin L. Masters [4], Alice Pébay [1,2], Sophie Chevalier[1,2], Jason Ha[1,2,5], Samantha Loi [6,7], Christopher J. Fowler[4], Christopher Rowe[8], Victor L. Villemagne[8], Edward N. Taylor[9], Christopher Fluke [10,11], Jean-Paul Soucy[12,13], Frédéric Lesage[14,15], Jean-Philippe Sylvestre[16], Pedro Rosa-Neto [12,17,18], Sulantha Mathotaarachchi[12,17], Serge Gauthier[18], Ziad S. Nasreddine[19], Jean Daniel Arbour[20], Marc-André Rhéaume[20], Sylvain Beaulieu[21], Mohamed Dirani[1,2,22], Christine T.O. Nguyen [3], Bang V. Bui[3], Robert Williamson[23], Jonathan G. Crowston[1,2] & Peter van Wijngaarden[1,2]

[1]Centre for Eye Research Australia, Royal Victorian Eye and Ear Hospital, East Melbourne, 3002 VIC, Australia. [2]Ophthalmology, Department of Surgery, University of Melbourne, Parkville, 3010 VIC, Australia. [3]Department of Optometry and Vision Sciences, University of Melbourne, Parkville, 3010 VIC, Australia. [4]The Florey Institute, The University of Melbourne, Parkville, 3010 VIC, Australia. [5]Faculty of Medicine, Nursing and Health Sciences, Monash University, Clayton, 3800 VIC, Australia. [6]Neuropsychiatry Unit, North Western Mental Health, Melbourne Health, Royal Melbourne Hospital, Parkville, 3050 VIC, Australia. [7]University of Melbourne, Department of Psychiatry, Parkville, 3010 VIC, Australia. [8]Austin

Health, Melbourne, 3084 VIC, Australia. [9]Centre for Astrophysics and Supercomputing, Swinburne University of Technology, Melbourne, 3122 VIC, Australia. [10]OzGrav-Swinburne, Centre for Astrophysics & Supercomputing, Swinburne University of Technology, Melbourne, 3122 VIC, Australia. [11]Advanced Visualisation Laboratory, Digital Research Innovation Capability Platform, Swinburne University of Technology, Melbourne, 3122 VIC, Australia. [12]McConnell Brain Imaging Centre, Montreal Neurological Institute, McGill University, Montreal, H3A 2B4 QC, Canada. [13]PERFORM Centre, Concordia University, Montreal, H4B 1R6 QC, Canada. [14]École Polytechnique de Montréal, Institut de génie biomédical, Département de Génie électrique, Montreal, H3C 3A7 QC, Canada. [15]Research Center, Montreal Heart Institute, Montreal, H1T 1C8 QC, Canada. [16]Optina Diagnostics, Montreal, H4T 1Z2 QC, Canada. [17]Translational Neuroimaging Laboratory, McGill Centre for Studies in Aging, Douglas Mental Health University Institute, Montreal, H4H 1R3 QC, Canada. [18]Alzheimer's Disease Research Unit, The McGill University Research Centre for Studies in Aging, McGill University, Montreal, H4H 1R3 QC, Canada. [19]MoCA Clinic and Institute, Greenfield Park, J4V 2J2 QC, Canada. [20]Clinique ophtalmologique 2121, Montreal, H3H 1G6 QC, Canada. [21]Département de médecine nucléaire, Hôpital Maisonneuve-Rosemont, Montreal, H1T 2M4 QC, Canada. [22]Singapore Eye Research Institute, Singapore National Eye Centre, Singapore, 169856, Singapore. [23]Murdoch Children's Research Institute and Department of Paediatrics, University of Melbourne, Melbourne, 3052 VIC, Australia

