## [Peer Review File · Nature Communications]

Reviewers' comments:

Reviewer #1 (Remarks to the Author):

The manuscript by Hadoux et al. provides the first evidence for the clinical utility of a non-invasive, label-free, live retinal hyperspectral (HS) imaging tool in humans to assess retinal reflectance score, which correlated with brain amyloid burden. The authors analyzed the spectral signature of retinal tissue – assessed by a previously developed retinal HS imaging approach – and explored the extent to which this correlates with brain amyloid-PET status. The manuscript reports several key findings: 1) the ability of retinal HS imaging to differentiate between individuals with high brain amyloid-PET load and controls; 2) A significant correlation between retinal HS score and brain amyloid status; 3) the reproducibility of results using a second cohort of subjects and HS imaging tools. This is a timely and clinically significant study demonstrating the potential utility of in vivo retinal HS imaging as a biomarker for AD. The authors present their results clearly, provide sufficient rationale, and effectively discuss the significance and limitations of their sophisticated and well-performed study. However, concerns exist regarding the fundamental nature and specificity of their retinal findings. At the minimum, authors should demonstrate in postmortem retinal tissues from AD/MCI patients compared to controls that accumulation of misfolded proteins (e.g. A β aggregates) could explain the observed variations in retinal HS signature, especially in the reported disease-significant regions F1 and S1. The findings could be strengthened by investigating the extent to which spectral signature in the retina represents A β pathology both in the retina and brain of MCI and AD cases, and whether longitudinal changes in retinal HS score correlate with disease progression.

Specific points:

- 1) Authors ought to be more careful in the use of amyloid vs. A β burden. The two terms are not always interchangeable and A β is a more specific term.
- 2) As expected, MMSE scores were lower in A β PET-positive cases; however, RNFL thickness was higher in A β PET-positive vs. -negative cases, which is contradictory to the vast majority of previous studies. These results and their implications should be discussed.
- 3) Based on the importance of RNFL thickness findings in AD and its previous correlations with MMSE scores, retinal HS data should be also presented with normalization to the RNFL thickness.
- 4) Does retinal vasculature impact retinal HS measurement? From the images it seems vasculature was excluded from regions of interest. Do retinal HS scores correlate with any of the retinal biomarkers previously published?
- 5) Generally, it is highly recommended to include key correlations between retinal HS scores and other demographic and disease-related metrics. These analyses could provide insights into whether retinal HS is affected by aging or other factors and if it can predict disease in the brain and/or cognitive function.
- 6) Authors indicate without citation(s) that similar to regional variation in retinal A β levels, retinal HS scores vary by region. By retinal A β levels, the authors mean retinal A β plaque burden? If yes, please specify and include the proper citation. It would be also important to, if possible, comparatively assess the regional variation of the two measures and to include a more in-depth discussion on this topic.
- 7) Background literature is somewhat missing. For example, the original manuscript identifying retinal A β deposits in MCI and AD patients is necessary to include.
- 8) Correlation data, especially between retinal HS scores and brain A β burden are intriguing but more discussion on their strength and biological interpretation could be included.
- 9) Finally, what does retinal HS score represent in the tissue? The authors do not pinpoint the underlying cause of variation in retinal spectra between A β -PET-positive and -negative subjects. Fundamentally, authors ought to demonstrate in patients' ex vivo tissues the co-presence of A β deposits (or other misfolded proteins or retinal abnormalities) with differential retinal HS scores (especially in the reported disease-affected regions, F1 and S1). This could provide an explanation for the variability in retinal HS signatures.

Reviewer #2 (Remarks to the Author):

The author's overall strategy appears to be to isolate components of spectral variability among subjects that potentially mask a spectral biomarker for AD. This is referred to as a correction for spectral variability. By use of a spectral classification method called DROP-D, a spectral model free of confounding variability is obtained, and HS scores for each class (or study group) are determined from the model. If I have understood correctly, the spectral representation of the disparity between model spectra (PET + and -) is explained using curve fits with an A β optical density spectrum recorded in vitro. Would authors please comment? I find that the methodology for much of the analytical methods is not explained in enough detail or demonstrated clearly in the captioned figures. A more rigorous presentation of analysis strategy and steps is needed. Often vague language is used to describe analysis steps and so it can be difficult to follow the work and have confidence about the authors' conclusions. Use of equations to document spectral normalisation and other areas of the analysis would help. On the other hand, presentations of study protocol, data collection, ethics, and cataract reach standards of technical and scientific writing.

Main concerns

The A β molecule has a broad absorption at 400 nm which is outside the range of wavelengths used. The authors' show the in vitro spectrum of light scatter of soluble A β . It is unlikely, given the differences in in vitro and in vivo environments, that the laboratory spectrum would match the scatter contribution in human recordings from the retina. However this spectrum could possibly act as a template for the in vivo case. Thus, although the authors' HS scores showed significant differences between study groups, pointing toward a biomarker, it is difficult to conclude from the curve fits that the spectral difference was due to retinal A β . To strengthen this part of the manuscript, would the authors please provide details about their curve fit, which parameters were used and allowed to change, and the concentration of the measured A β . Is 1 mg/ml the solution concentration used?

An additional caution is that near the start of analysis the authors apparently normalised reflectance spectra from each retinal area using the average spectrum in order to highlight the degree of intersubject variability (in lines 141-144). In fact, in the human population there is large variation in retinal melanin level that needs to be taken into account if spectral biomarkers are to be obtained from retinal data. However, it is likely that the AD biomarker is present in these average spectra. If separate averaged spectra from PET + and - groups were used to normalise within each group, the presence of the desired marker could have been partially erased by this normalisation. Please comment on this possibility.

Minor concerns

Usually the label on the plot axis is written as the name of the variable with units shown in parentheses. Author's have used a label of Reflectance (Log) on several plots. Suggest relabeling as Log Reflectance or Ln Reflectance, and add (%) if reflectance is expressed as a percentage.

Fig. S1. Control curves are barely visible, with some separation of the control and case curves near 550 nm. Is this due to the control and case curves exactly overlapping or is it because error bars mask? Interesting that the error bars appear large within the dynamic range of the reflectance. Error bars should be labeled in Figure as to type (SE, SD, ..).

Fig S3. More explanation needed about the decision that removal of two components is optimal.

Fig S5. What were the linear combinations of fundus spectral components that produced W1 and W2. Please give their respective contributions.

Fig S8. How was the conclusion for good agreement reached? The curves show coherence over only limited wavelength ranges and are closest in the infrared near 750 nm. Is this likely due to lower absorptions by pigments and hemoglobin, or would the spectrum of the light source explain it?

Fig S9. Unfortunately the caption doesn't help in understanding this effect of removal. The analysis done here is complex. Throughout the manuscript, please use figure captions as much as possible to buttress the arguments in text, and explain how to interpret the relationship between the curves chosen for the plot.

Response to Reviewers' comments:

We thank the reviewers for their careful appraisal of our manuscript. In light of these comments we have undertaken additional preclinical experiments and have made major revisions to our manuscript. Major changes include the addition of mouse *in vivo* hyperspectral imaging findings and a revised simulation of the retinal spectral effect of A β , which now appears in the supplementary materials. We have also added a detailed account of the data processing methods (supplementary materials). Our manuscript is significantly stronger for these additions and amendments.

We have provided the comments of each reviewer below and our responses in turn. Our responses highlight and contextualise the amendments that we have made to the manuscript.

Reviewer #1:

The manuscript by Hadoux et al. provides the first evidence for the clinical utility of a non-invasive, label-free, live retinal hyperspectral (HS) imaging tool in humans to assess retinal reflectance score, which correlated with brain amyloid burden. The authors analyzed the spectral signature of retinal tissue – assessed by a previously developed retinal HS imaging approach – and explored the extent to which this correlates with brain amyloid-PET status. The manuscript reports several key findings: 1) the ability of retinal HS imaging to differentiate between individuals with high brain amyloid-PET load and controls; 2) A significant correlation between retinal HS score and brain amyloid status; 3) the reproducibility of results using a second cohort of subjects and HS imaging tools. This is a timely and clinically significant study demonstrating the potential utility of *in vivo* retinal HS imaging as a biomarker for AD. The authors present their results clearly, provide sufficient rationale, and effectively discuss the significance and limitations of their sophisticated and well-performed study. However, concerns exist regarding the fundamental nature and specificity of their retinal findings. At the minimum, authors should demonstrate in postmortem retinal tissues from AD/MCI patients compared to controls that accumulation of misfolded proteins (e.g. A β aggregates) could explain the observed variations in retinal HS signature, especially in the reported disease-significant regions F1 and S1. The findings could be strengthened by investigating the extent to which spectral signature in the retina represents A β pathology both in the retina and brain of MCI and AD cases, and whether longitudinal changes in retinal HS score correlate with disease progression.

Introductory comments

Reviewer's comment	Response
(1) However, concerns exist regarding the fundamental nature and specificity of their retinal findings. At the minimum, authors should demonstrate in postmortem retinal tissues from AD/MCI patients compared to controls that accumulation of misfolded proteins (e.g. A β aggregates) could explain the observed variations in retinal HS signature, especially in the reported disease-significant regions F1 and S1. The findings could be strengthened by investigating the extent to which spectral signature in the retina represents A β pathology both in the retina and brain of MCI and AD cases, and whether longitudinal changes in retinal HS score correlate with disease progression.	We have performed hyperspectral imaging in post-mortem eyes of two subjects with histopathologically confirmed AD (brain studies) and despite a short post-mortem interval (less than 6 hours) corneal opacification (due to oedema) precluded high quality imaging. Furthermore, the assumptions underlying our data processing method no longer apply in the absence of ocular perfusion. To redesign an imaging system (a microscope as opposed to a fundus camera) and algorithm for ex vivo retinal tissue analysis would be to recapitulate the work reported by Moore and Vince (1). The post-mortem human retinal studies of Koronyo and colleagues (2, 3) identify retinal accumulation of Aβ as well as regional variations in Aβ load, with more extensive deposits of Aβ plaques in the superior retina. In order to address concerns regarding the fundamental nature and specificity of the findings (i.e., that the measured spectral signature actually corresponds to retinal Aβ) we have undertaken extensive additional preclinical studies (see updated manuscript Method and Results section). In brief, we have adapted a hyperspectral retinal camera for in vivo imaging of mouse eyes. We have imaged 5xFAD mice that are known to accumulate Aβ in retina and brain and matched controls. We have recalibrated our human spectral model (450-900nm) to match spectral range used in the mouse imaging study (450-680nm) and calculated HS scores for each eye using the identical processing method that was applied to images acquired from our human subjects. We show a

	significant group difference between 5xFAD and control mice ($p = 0.03$). Moreover, the human model classified individual mice into high and low Aβ groups with high accuracy (AUC = 0.82, Fig. 7). As the primary difference between these matched inbred mouse strains is over expression of Aβ, both in brain and in retina, this finding adds further support to our claim that the retinal spectral signature is due to Aβ. We agree that it would be extremely valuable to document longitudinal changes in retinal Aβ load via HS imaging and correlating this with brain load, however we assert that this is the subject of a separate study, owing to the fact that this is a major undertaking (in terms of cost and time). We have embarked on a human clinical study to address this question with the expectation of having preliminary results in 3-5 years. We further assert that the body of work presented in our revised manuscript constitutes an important contribution to the field in its own right.
--	---

Specific points:

Reviewer's comment	Response
(1) Authors ought to be more careful in the use of amyloid vs. Ab burden. The two terms are not always interchangeable and Ab is a more specific term.	We have replaced all usages of the term "amyloid burden" and replaced this with "A β burden". We now only make reference to the term "burden" in the context of brain A β PET findings.
(2) As expected, MMSE scores were lower in Ab PET-positive cases; however, RNFL thickness was higher in Ab PET-positive vs. -negative cases, which is contradictory to the vast majority of previous studies. These results and their implications should be discussed.	We found no statistically significant difference in retinal nerve fibre layer (RNFL) thickness between groups ($p = 0.3$). Whilst there have been published reports of an association between RNFL thinning and both AD and non-AD associated cognitive decline these are weak associations and they are not predictive at the individual level. A metanalysis on this subject concluded "RNFL thickness cannot be used as a biomarker of AD" (4). We have added the following comment to the discussion to clarify that we found no difference in RNFL thickness between groups in the present study and to contextualise the utility of OCT as a biomarker of AD: "...a recent large longitudinal study found that subtle changes in thickness of the retinal nerve fibre layer, measured using optical coherence tomography (OCT), were associated with cognitive decline (33). It is possible that these and related structural alterations in the retina in AD may contribute to the detected spectral signature, however no such changes were detected using OCT in the present study. Whilst OCT studies have identified associations between retinal nerve fibre layer thinning and AD, these effects are small in magnitude, non-specific and not predictive at the individual level." Using the data derived from this meta-analysis (N = 1751), the average RNFL thickness difference between AD and controls was 9.7 μm and the pooled standard deviation was 22.6 μm. The power to detect a significant difference in our study with approximately 40 participants would only be 0.27. Accordingly, our study is underpowered for the detection of an RNFL difference between groups.
(3) Based on the importance of RNFL thickness findings in	As highlighted above, we did not find a statistically

AD and its previous correlations with MMSE scores, retinal HS data should be also presented with normalization to the RNFL thickness.	significant difference in RNFL thickness between the two groups. For illustrative purposes a normalisation of the HS scores for RNFL thickness is provided below. The findings are similar to that for non-normalised data and are therefore not included in the manuscript.  A 0.004 0.002 0 -0.002 -0.004 HS score / RNFL p = 0.002 PET- PET+
(4) Does retinal vasculature impact retinal HS measurement? From the images it seems vasculature was excluded from regions of interest. Do retinal HS scores correlate with any of the retinal biomarkers previously published?	As noted by the reviewer, retinal blood vessels were excluded from the analysis. Blood vessels have strong spectral signatures, and this is the basis of spectral oximetry analysis used by others. Given that our method involves the semi-automated sampling of defined regions of interest (ROI) and given that retinal vascular anatomy varies considerably between individuals, each ROI may contain varying numbers of pixels overlying blood vessels. Inclusion of the retinal blood vessels therefore results in unnecessary variability in the spectral profile. We have now clarified this in the Methods section of the manuscript: “Given that retinal vascular anatomy varies considerably between individuals, each sampling location may contain varying numbers of pixels overlying blood vessels. Retinal blood vessels have a prominent spectral signature and thus failure to exclude vessels results in unnecessary variability in the spectral profile.” We have not assessed correlation of our HS scores with other retinal biomarkers of AD. Retinal photographic vascular biomarkers correlate poorly with AD and with brain PET Aβ levels (5). We do not have access to OCT-angiography and therefore cannot correlate our scores with perifoveal retinal vascular changes (6). As described in the response to Reviewer #1 specific point (2) above we did not find a correlation between RNFL thickness, nor any other OCT parameter, and our HS score, or with brain PET Aβ status.
(5) Generally, it is highly recommended to include key correlations between retinal HS scores and other demographic and disease-related metrics. These analyses could provide insights into whether retinal HS is affected by aging or other factors and if it can predict disease in the brain and/or cognitive function.	We thank the reviewer for this important comment. We now report correlation between HS score and age, MMSE and RNFL thickness in the revised manuscript as follows: “No significant correlations were found between retinal HS score and age ($r = 0.18$, $p = 0.3$, 95% CI: $-0.17 - 0.48$), MMSE ($r = -0.29$, $p = 0.1$, 95% CI: $-0.57 - 0.05$) or RNFL thickness ($r = 0.15$, $p = 0.4$, 95% CI: $-0.2 - 0.48$).”
(6) Authors indicate without citation(s) that similar to regional variation in retinal Ab levels, retinal HS scores vary by region. By retinal Ab levels, the authors mean	The reviewer is correct – the limited published data identify a difference in distribution of plaques but not of

retinal Ab plaque burden? If yes, please specify and include the proper citation. It would be also important to, if possible, comparatively assess the regional variation of the two measures and to include a more in-depth discussion on this topic.	soluble or oligomeric Aβ – largely as analyses have centred on immunohistochemical studies of retinal tissue, rather than quantitative assays by region. We have shown that the spectral signature difference between PET+ cases and PET- controls is most marked in the superior part of the posterior pole of the retina, but it remains significantly different when all of the 6 sampling locations are considered together (Figure 4). We have removed the following statement that appeared in the previous version of the manuscript: “Consistent with post-mortem histopathology which has demonstrated regional variation in Ab levels, retinal HS scores also varied by region.” We have qualified our claim that Aβ levels appear to vary by region in the human retina in the revised manuscript as follows: “HS scores were highest in the superior (S1) and foveal (F1) sampling locations. Little is known of regional variations in the distribution of Aβ in the human retina. One human study has identified increased Aβ plaque distribution in the superior retina, however the generalisability of this finding and its pathophysiological basis is not known.”
(7) Background literature is somewhat missing. For example, the original manuscript identifying retinal Ab deposits in MCI and AD patients is necessary to include.	We thank the reviewer for noticing this omission. We have added the reference to Koronyo et al that was missing.
(8) Correlation data, especially between retinal HS scores and brain Ab burden are intriguing but more discussion on their strength and biological interpretation could be included.	We have amended the wording of the manuscript to highlight that this is a moderate positive correlation. We also emphasise that this was a post-hoc analysis and that the data analysis method was not optimised for quantitative associations between the retinal HS score and brain Aβ PET burden. We also add the following qualifying statement: “Whilst the latter finding is interesting, further studies are needed to establish the extent to which retinal HS scores are quantitatively associated with brain Aβ levels.”
(9) Finally, what does retinal HS score represent in the tissue? The authors do not pinpoint the underlying cause of variation in retinal spectra between Ab-PET-positive and -negative subjects. Fundamentally, authors ought to demonstrate in patients’ ex vivo tissues the co-presence of Aβ deposits (or other misfolded proteins or retinal abnormalities) with differential retinal HS scores (especially in the reported disease-affected regions, F1 and S1). This could provide an explanation for the variability in retinal HS signatures.	We thank the reviewer for this astute comment. We have devoted considerable time to additional preclinical studies to address this point. Please refer to our response to Reviewer #1 introductory comment (1) for a description of this work. Our manuscript is now significantly stronger as a result of these additions.

Reviewer #2:

The author's overall strategy appears to be to isolate components of spectral variability among subjects that potentially mask a spectral biomarker for AD. This is referred to as a correction for spectral variability. By use of a spectral classification method called DROP-D, a spectral model free of confounding variability is obtained, and HS scores for each class (or study group) are determined from the model. If I have understood correctly, the spectral representation of the disparity between model spectra (PET + and -) is explained using curve fits with an A β optical density spectrum recorded in vitro. Would authors please comment? I find that the methodology for much of the analytical methods is not explained in enough detail or demonstrated clearly in the captioned figures. A more rigorous presentation of analysis strategy and steps is needed. Often vague language is used to describe analysis steps and so it can be difficult to follow the work and have confidence about the authors' conclusions. Use of equations to document spectral normalisation and other areas of the analysis would help. On the other hand, presentations of study protocol, data collection, ethics, and cataract reach standards of technical and scientific writing.

Introductory comments

Reviewer's comment	Response
(1) If I have understood correctly, the spectral representation of the disparity between model spectra (PET + and -) is explained using curve fits with an A β optical density spectrum recorded in vitro. Would authors please comment?	The reviewer is correct. We have performed a simulation to ascertain whether the observed disparity between model spectra (PET + and PET -) could be attributable to Aβ. To do so we have used the optical density spectrum of human Aβ in solution in addition to known reflectance profiles of the ocular components that have major influences on the reflectance of light. We acknowledge that this is indirect evidence in support of our findings. Accordingly, we have opted to move these data to the Supplementary materials (Supplementary Methods 2, Supplementary Results 1; Fig. S10 & S11, Table S8 & S9). We also acknowledge the convoluted way we in which previously attempted to demonstrate the importance of the Aβ spectral profile to explain the model. We have substantially altered our simulation, using only conventional statistical methods, and the description of our approach is now much clearer. We consider that these amendments have substantially strengthened our manuscript. As outlined in our response to Reviewer #1 introductory comment (1), our attempts at correlating the measured spectral signal with the quantity of retinal Aβ in human retina have been ineffective due to challenges of performing HS retinal imaging in the post-mortem eye.
(2) The methodology for much of the analytical methods is not explained in enough detail or demonstrated clearly in the captioned figures	We have provided more detailed descriptions of our analytical methods throughout the revised manuscript. Examples include the comprehensive description of the DROP-D method (Supplementary Methods 1) and the complete revision of the simulation of the spectral influence of A β (see Response to Reviewer #1 introductory comment (1) above).
(3) Use of equations to document spectral normalisation and other areas of the analysis would help.	This comment is well received. We have added a comprehensive description of the DROP-D method to the revised manuscript (Supplementary Methods 1). The normalisation method is now clearly explained in the Fig. 2 caption: "...Normalised spectra were obtained by dividing each spectrum by the average spectrum of all the participants in the principal cohort (PET+ and PET-)"

Main concerns

Reviewer's comment	Response
(1) The Aβ molecule has a broad absorption at 400 nm which is outside the range of wavelengths used. The authors' show the in vitro spectrum of light scatter of soluble Aβ. It is unlikely, given the differences in in vitro and in vivo environments, that the laboratory spectrum would match the scatter contribution in human recordings from the retina. However this spectrum could possibly act as a template for the in vivo case. Thus, although the authors' HS scores showed significant differences between study groups, pointing toward a biomarker, it is difficult to conclude from the curve fits that the spectral difference was due to retinal Aβ. To strengthen this part of the manuscript, would the authors please provide details about their curve fit, which parameters were used and allowed to change, and the concentration of the measured Aβ. Is 1 mg/ml the solution concentration used?	We agree with the reviewer that using the ex vivo spectrum is better used as a template for curve fitting than for direct comparison of spectroscopic optical density. The curve fitting approach has been completely updated and moved to the Supplementary materials section of the manuscript (please see Supplementary Methods 2, Supplementary Results 1, Fig. S9 & S10, Table S8 & S9). 1 mg/ml solution concentration was used. This is now clearly stated in the manuscript. Please also note that we have performed additional mouse hyperspectral imaging studies to further validate our findings in humans (please refer to our response to Reviewer #1 introductory comment (1)).
(2) An additional caution is that near the start of analysis the authors apparently normalised reflectance spectra from each retinal area using the average spectrum in order to highlight the degree of intersubject variability (in lines 141-144). In fact, in the human population there is large variation in retinal melanin level that needs to be taken into account if spectral biomarkers are to be obtained from retinal data. However, it is likely that the AD biomarker is present in these average spectra. If separate averaged spectra from PET + and - groups were used to normalise within each group, the presence of the desired marker could have been partially erased by this normalisation. Please comment on this possibility.	We agree that if we had used the average spectrum per group for normalisation this could have diminished or removed the biomarker signal. Instead we used the overall average spectrum of all our participants to perform the normalisation, in order to avoid this problem. We thank the reviewer for highlighting the fact that this was not clearly stated in the manuscript. The revised manuscript has been modified as follows: "Reflectance spectra normalised to the average spectrum of all the participants in the principal cohort (PET+ and PET-) are displayed [...]." and in Fig 2 caption: "[...] Normalised spectra were obtained by dividing each spectrum by the average spectrum of all the participants in the principal cohort (PET+ and PET-)."

Minor concerns

(1) Usually the label on the plot axis is written as the name of the variable with units shown in parens. Author's have used a label of Reflectance (Log) on several plots. Suggest relabeling as Log Reflectance or Ln Reflectance, and add (%) if reflectance is expressed as a percentage.	Thank you, we have corrected all instances of this labelling in the manuscript. Labels now appear as Log Reflectance or Normalised Log Reflectance.
(2) Fig. S1. Control curves are barely visible, with some separation of the control and case curves near 550 nm. Is this due to the control and case curves exactly overlapping or is it because error bars mask? Interesting that the error bars appear large within the dynamic range of the reflectance. Error bars should be labeled in Figure as to type (SE, SD, ..).	The reviewer is correct, error bars were shown as SD in this figure and as SEM in the other figures, which is confusing. SEM is now used for all figures and each figure legend now defines the error bars used. As noted by the reviewer, the largest difference between case and control curves is near 550 nm. 2-fold variation in the dynamic range across the wavelengths prevents visualisation of the small difference between groups. Accordingly, normalised data are also provided in Figure 2. We have included a statement in the revised manuscript to make this clear: "...due to the large dynamic range of fundus reflectance (approximately 2 orders of magnitude from 450 – 900 nm), raw reflectance spectra are not useful for visualization of group differences (Fig. S1A-F)"
(3) Fig S3. More explanation needed about the decision that removal of two components is optimal.	The caption for Figure S3 further explains the justification for the removal of 2 components:

	“The optimal number of components was selected to be at the overall maximum across each sampling location. Selecting more (or fewer) components would decrease the classification results on average across all sampling locations.”
(4) Fig S5. What were the linear combinations of fundus spectral components that produced W1 and W2. Please give their respective contributions.	Thank you for this valuable suggestion. Supplementary tables (Tables S1 to S6) were added to provide the respective contribution to W1 and W2 for each sampling location.
(5) Fig S8. How was the conclusion for good agreement reached? The curves show coherence over only limited wavelength ranges and are closest in the infrared near 750 nm. Is this likely due to lower absorptions by pigments and hemoglobin, or would the spectrum of the light source explain it?	The conclusion that there is good agreement is based on correspondence between the location and respective amplitude of the local minima and maxima for all sampling locations (i.e., 450, 550 and 750 nm). We have amended the caption for Figure S8 to make this clear: “Good agreement is characterised by having local extrema (wavelength and amplitude) which are similar across sampling locations (i.e., 450, 550 and 750 nm).” The similarity between curves indicates that the same spectral confounders need to be removed in order to differentiate PET+ from PET- participants, irrespective of the retinal sampling location (Fig. S5, Table S1 to S6).
(6) Fig S9. Unfortunately the caption doesn't help in understanding this effect of removal. The analysis done here is complex. Throughout the manuscript, please use figure captions as much as possible to buttress the arguments in text, and explain how to interpret the relationship between the curves chosen for the plot.	We thank the reviewer for this helpful suggestion. We have substantially revised the manuscript and have improved the figure captions. We agree that Figure S9 in the previous version of the manuscript was confusing. We have substantially revised our simulation and statistical analysis as described above (Reviewer #1 introductory comment (1)) and in the Supplementary Materials (Supplementary Methods 2, Supplementary Results 1, Fig. S10 & S11, Table S8 & S9) .

References

1. S. S. More, R. Vince, Hyperspectral imaging signatures detect amyloidopathy in Alzheimer's mouse retina well before onset of cognitive decline, *ACS Chem Neurosci* **6**, 306–315 (2015).
2. M. Koronyo-Hamaoui, Y. Koronyo, A. V. Ljubimov, C. A. Miller, M. H. K. Ko, K. L. Black, M. Schwartz, D. L. Farkas, Identification of amyloid plaques in retinas from Alzheimer's patients and noninvasive in vivo optical imaging of retinal plaques in a mouse model, *NeuroImage* **54**, S204–17 (2011).
3. Y. Koronyo, D. Biggs, E. Barron, D. S. Boyer, J. A. Pearlman, W. J. Au, S. J. Kile, A. Blanco, D.-T. Fuchs, A. Ashfaq, S. Frautschy, G. M. Cole, C. A. Miller, D. R. Hinton, S. R. Verdooner, K. L. Black, M. Koronyo-Hamaoui, Retinal amyloid pathology and proof-of-concept imaging trial in Alzheimer's disease, *JCI Insight* **2**, 1–19 (2017).
4. J. den Haan, F. D. Verbraak, P. J. Visser, F. H. Bouwman, Retinal thickness in Alzheimer's disease: A systematic review and meta-analysis, *Alzheimers Dement (Amst)* **6**, 162–170 (2017).
5. C. Y.-L. Cheung, Y.-T. Ong, M. K. Ikram, C. Chen, T. Y. Wong, J. C. de la Torre, Ed. Retinal Microvasculature in Alzheimer's Disease, *Journal of Alzheimer's Disease* **42**, S339–S352 (2014).
6. S. P. Yoon, D. S. Grewal, A. C. Thompson, B. W. Polascik, C. Dunn, J. R. Burke, S. Fekrat, Retinal Microvascular and Neurodegenerative Changes in Alzheimer's Disease and Mild Cognitive Impairment Compared with Control Participants, *Ophthalmology Retina* (2019), doi:10.1016/j.oret.2019.02.002.

Reviewers' comments:

Reviewer #1 (Remarks to the Author):

Hadoux and colleagues addressed nicely the previous comments. Minor concerns remain:

1. Abstract: For accuracy, the manuscript could benefit from toning down the statement "In keeping with this, we have identified a retinal imaging biomarker of brain Ab burden using hyperspectral (HS) imaging." Perhaps using ...we explored a possible retinal...is more appropriate.
2. Results: The addition of 5xFAD mouse HS imaging is highly advantageous. To validate increased HS score due to Abeta content in the retina, it is recommended to perform Abeta histology on retinal tissues from mice that underwent HS imaging and compare the histological signal to the in vivo HS score.
3. Introduction (second paragraph): Since the authors mention the studies showing accumulation of retinal Abeta in both in human patients and in mouse models, many references are missing including the original identification manuscripts both in humans and in transgenic mouse models of AD. In general, more background literature is needed for in vivo and ex vivo data on Abeta in the retina.
4. Introduction (last paragraph): "Hyperspectral imaging was pioneered by scientists at NASA for remote sensing of the earth from satellites (22) and has been used extensively in agriculture (23), food processing (24), mineralogy (25) and more recently for medical applications (26)," while interesting, sentence is not necessary for the scope of this manuscript.
5. How would the HS signal appear in other neurodegenerative eye disease such as glaucoma and/or AMD? It would be important if this was included or referenced.
6. Are there any disparities between the left and right retinal HS scores? The extent of intrasubject variability between retinal regions and different eyes is of interest, and it would be great if authors can comment on the feasibility of obtaining HS signals from mid or far-peripheral regions.
7. Discussion: Beyond Abeta accumulation, another possible cause of increased HS score in the retina of PET+ subject could be heightened inflammation.

Reviewer #2 (Remarks to the Author):

The revised manuscript is much improved regarding readability, however there is still concern about the overall strategy. The description of statistical methods for evaluating retinal spectral components, including the introduction to Drop D methods, are now easier to understand, although only a fraction of readers will follow these in detail. For the new section, mouse HS recordings, please describe the resampling from 320 - 680 / 10 nm (1 nm increments) with xenon to 450 - 680 / 5 nm data for hyperspectral modeling, and the vessel segmentation using 390 - 460 nm averaged images. Would it be technically feasible with your HS system to determine more than a single reflectance spectrum from mouse retina? Or was there no need for this, considering that the goal with mouse was simply to corroborate human results with an established retinal model for A β burden? Followup recordings using the fellow with a second instrument showed that the HS retinal images came from reliable recording technique, however if performed on the same eye this reviewer feels the test would be more conclusive.

The authors have developed an elegant and complex spectral analysis to characterize spectral features of retinal structure that they believe mask a spectral biomarker for AD. This work was undertaken after not finding significant spectral differences (case vs control) in either reflectance or normalised reflectance spectra. These procedures may in fact be necessary to detect the biomarker. This reviewer believes, on the basis of data shown for the normalised version, that there is a significant PET + spectral component in the authors' recordings that is not in the control, and that this could be detected in some form of corrected spectrum without need to mitigate the complex spectral features from retinal structures or components. It is still not confirmed that this is an amyloid biomarker. The authors state that the large dynamic range of spectral intensities (light intensity as a function of wavelength over vis and near ir) in the reflectance spectra from different regions or retina caused these to be unsuited for the visualization of group differences.

Fig. S1 shows that test and control spectra from the six retinal regions, which are plotted on compressed log scales, largely overlap. Accordingly, statistical tests did not produce a significant difference between cases (although $p < 0.05$ was shown for a few wavelengths in area S1, Figure S2). The p values approached significance at wavelengths near hemoglobin peaks, which seems to indicate that inter-subject variation in a retinal component, namely hemoglobin, is cause for most of the difference. This reviewer agrees that the case and control curves do not reveal spectral change that is dependent on case. The authors then performed a correction on the raw spectra by normalisation, using the average raw retinal spectrum from all subjects (PET + and PET -). This procedure takes out the large dynamic range and cancels effects from retinal features that are in common (eg. hemoglobin variations will average and cancel). The more subtle spectral features are now visualizable. Apparently the correction is a necessary step for seeing case-related differences of size well below the dynamic range of retinal reflectance spectra. In the main text the normalised spectra are said to highlight variations from retinal structures at the different locations, and also the large degree of inter-subject variability using uncorrected spectral data (also stated in Figure 2F-K caption). Actually, the SE (error bars) shown in the figure involve the variance of each case spectrum and the average spectrum, and would be determined using error propagation. These new errors would not be exclusive to original variability in the uncorrected spectra. The variability shown on corrected spectra is large compared to changes in mean values about zero or between the test and control spectrum. Some features of the corrected spectra do in fact pertain to retinal structure variability, however the spectra also show a distinct feature related solely to case. Please note that Fig2F-K shows a consistent intensity inversion across wavelengths between the test and control spectra, more so for locations F1 and F2. The curves are flipped. Normalisation using the full average from both groups (PET + and PET -) has produced opposing effects for each group. This is not mentioned by the authors. Such a trend in the corrected spectra would not come from variation in retinal structure so much as from a difference between test and control. If a spectral component from an AD biomarker were present in just one of the groups, normalization by an average taken over case and control would act, by cancellation, to leave a positive residual spectrum of the marker in one group, and a negative residual in the other. Spectra in Fig2F-K are consistent with this possibility. This reviewer believes there is evidence for a biomarker in the authors' data and that it can be recovered without mitigation for effects of retinal structures on retinal spectra. Nonetheless, these more advanced analyses may be of interest to certain readers and are explained in a way these readers would appreciate.

Response to Reviewers' comments:

We thank the reviewers for their careful appraisal of our manuscript. In light of these comments we have undertaken additional preclinical experiments and have made revisions to our manuscript. These include the addition of amyloid beta immunohistochemistry to supplement the mouse in vivo hyperspectral imaging findings. We have further clarified our data processing methods (supplementary materials). Our manuscript is stronger for these additions and amendments.

We have provided the comments of each reviewer below and our responses in turn.

Reviewer #1:

Hadoux and colleagues addressed nicely the previous comments. Minor concerns remain:

Reviewer's comment	Response
(1) Abstract: For accuracy, the manuscript could benefit from toning down the statement "In keeping with this, we have identified a retinal imaging biomarker of brain Ab burden using hyperspectral (HS) imaging." Perhaps using ...we explored a possible retinal...is more appropriate.	We appreciate this feedback and we have modified the abstract as follows: "As Aβ has a wavelength-dependent effect on light scatter, we investigated the potential for in vivo retinal hyperspectral (HS) imaging to serve as a biomarker of brain Aβ."
(2) Results: The addition of 5xFAD mouse HS imaging is highly advantageous. To validate increased HS score due to Abeta content in the retina, it is recommended to perform Abeta histology on retinal tissues from mice that underwent HS imaging and compare the histological signal to the in vivo HS score.	Retinal immunohistochemistry for amyloid beta was performed on mice that underwent antemortem in vivo hyperspectral retinal imaging (three 5xFAD mice and one aged-matched control mouse). These experiments confirm the presence of retinal amyloid beta plaques in 5xFAD mice that are not present in control mice (L345-364 & Fig 7). This is consistent with the findings of others (Alexandrov et al., Neuroreport 2011).
(3) Introduction (second paragraph): Since the authors mention the studies showing accumulation of retinal Abeta in both in human patients and in mouse models, many references are missing including the original identification manuscripts both in humans and in transgenic mouse models of AD. In general, more background literature is needed for in vivo and ex vivo data on Abeta in the retina.	Additional references have been added to the introduction.
(4) Introduction (last paragraph): "Hyperspectral imaging was pioneered by scientists at NASA for remote sensing of the earth from satellites (22) and has been used extensively in agriculture (23), food processing (24), mineralogy (25) and more recently for medical applications (26)," while interesting, sentence is not necessary for the scope of this manuscript.	This sentence was removed.
(5) How would the HS signal appear in other neurodegenerative eye disease such as glaucoma and/or AMD? It would be important if this was included or referenced.	As presented in this revised version of the manuscript, the retinal nerve fibre layer (RNFL) thickness, an objective structural measure of glaucoma, was not significantly correlated with the HS score. (L 227) Our study group did not include participants with late age-related macular degeneration (AMD). We performed a post-hoc analysis to evaluate the influence of drusen, a biomarker of early and intermediate AMD, on HS scores. The presence of drusen was not correlated with HS scores in either group (PET + cases and PET-

	controls). This additional finding now appears in the revised manuscript as follows (L 228): “HS scores were not significantly influenced by the presence of drusen in the case ($p = 0.30$, 95% CI: $-0.15 - 0.05$) or control groups ($p = 0.51$, 95% CI: $-0.19 - 0.10$).”
(6) Are there any disparities between the left and right retinal HS scores? The extent of intrasubject variability between retinal regions and different eyes is of interest, and it would be great if authors can comment on the feasibility of obtaining HS signals from mid or far-peripheral regions.	As shown in Figure 6, the HS scores obtained from left and right eyes are significantly correlated ($r = 0.6$, $p = 0.0002$). The HS camera used in this study is optimised for imaging of the posterior pole with a 30 degree of field of view. Accordingly, the imaging protocol used for all study participants involved posterior pole image acquisition. In our recent imaging studies we have demonstrated that it is possible to obtain high quality images of the mid-periphery (up to 60 degrees of combined field of view) with cooperative and well dilated (>6mm) participants.
(7) Discussion: Beyond Abeta accumulation, another possible cause of increased HS score in the retina of PET+ subject could be heightened inflammation.	We have included inflammation as another plausible cause of an increased HS score in the retina as follows below. We also include a number of key citations that identify inflammatory changes in the retina in AD. “It is possible that the spectral effect observed in this study was due to other factors that are associated with brain PET Aβ status, such as iron accumulation, tau phosphorylation or inflammation³¹⁻³³.”

Reviewer #2:

The revised manuscript is much improved regarding readability, however there is still concern about the overall strategy.

Reviewer's comment	Response
(1) The description of statistical methods for evaluating retinal spectral components, including the introduction to Drop D methods, are now easier to understand, although only a fraction of readers will follow these in detail.	We appreciate this positive feedback.
(2) For the new section, mouse HS recordings, please describe the resampling from 320 - 680 / 10 nm (1 nm increments) with xenon to 450 - 680 / 5 nm data for hyperspectral modeling, and the vessel segmentation using 390 - 460 nm averaged images.	The methods describing the mouse experiment have been updated as follows: “Blood vessels visible in the inner retina were automatically segmented using an average image from wavelengths 390 to 460 nm. The mask used to remove blood vessels was derived using a series of difference of Gaussian (DoG) (3σ) calculated using variance parameters 1 to 5 (0.2 step) for the small and 4 to 14 (1 step) for the large filters.” and, “The reflectance spectra were resampled between 450 and 680 nm in 5 nm steps using a linear interpolation method.”
(3) Would it be technically feasible with your HS system to determine more than a single reflectance spectrum from mouse retina? Or was there no need for this, considering that the goal with mouse was simply to corroborate human results with an established retinal model for A β burden?	The reviewer is correct, while it is possible to extract different ROI in the mouse retina, the aim of the animal experiment was only to corroborate the human findings with an established transgenic mouse model that is known to accumulate A β in the retina.
(4) Followup recordings using the fellow with a second instrument showed that the HS retinal images came from reliable recording technique, however if performed on the same eye this reviewer feels the test would be more conclusive.	We thank the reviewer for this insightful comment and we thus added a supplementary figure corresponding to intra-session repeatability as 25 participants had repeat images acquired during the imaging session. The test-retest performance is provided as a correlation and was >0.83 for each sampling location: “Intra-session repeatability of HS score is high Repeat retinal HS images in the same imaging session by the same operator were captured for 25 participants in order to assess intra-session repeatability. HS scores were computed for each location and the correlation between repeat images was calculated. HS scores were strongly correlated at all sampling locations (Fig. S10).”
(5) The authors have developed an elegant and complex spectral analysis to characterize spectral features of retinal structure that they believe mask a spectral biomarker for AD. This work was undertaken after not finding significant spectral differences (case vs control) in either reflectance or normalised reflectance spectra. These procedures may in fact be necessary to detect the biomarker. This reviewer believes, on the basis of data shown for the normalised version, that there is a significant PET + spectral component in the authors' recordings that is not in the control, and	We thank the reviewer for this supportive comment.

that this could be detected in some form of corrected spectrum without need to mitigate the complex spectral features from retinal structures or components.	
(6) It is still not confirmed that this is an amyloid biomarker.	The reviewer is correct. We have provided several lines of evidence that associate the observed spectral effect with retinal amyloid beta, but we have acknowledged that we do not have conclusive proof of this: “We do not have conclusive evidence that the HS score is due to retinal Aβ alone. It is possible that the spectral effect observed in this study was due to other factors that are associated with brain PET Aβ status, such as iron accumulation, tau phosphorylation or inflammation³¹⁻³³.”
(7) The authors state that the large dynamic range of spectral intensities (light intensity as a function of wavelength over vis and near ir) in the reflectance spectra from different regions or retina caused these to be unsuited for the visualization of group differences. Fig. S1 shows that test and control spectra from the six retinal regions, which are plotted on compressed log scales, largely overlap. Accordingly, statistical tests did not produce a significant difference between cases (although $p < 0.05$ was shown for a few wavelengths in area S1, Figure S2). The p values approached significance at wavelengths near hemoglobin peaks, which seems to indicate that inter-subject variation in a retinal component, namely hemoglobin, is cause for most of the difference. This reviewer agrees that the case and control curves do not reveal spectral change that is dependent on case.	The reviewer is correct in that the large dynamic range is problematic for visualisation in a figure. We took a conservative approach with significance level and corrected for false discovery rate (for each wavelength tested), which means that the alpha level of significance is lower than 0.05 and therefore no uncorrected wavelength was significant. We believe that the near significance at 550 nm is not due to a variation in haemoglobin but instead corresponds to a spectral region where there is the least amount of spectral variability within groups. Below 550 nm there is a strong variability in ocular media and in macular pigment in the retina which masks the spectral effect of amyloid beta. At the longer wavelengths, the light is able to penetrate more deeply into the tissue where the main driver of reflectance is the choroid (blood and melanin). To validate this statement, we have now added a standard deviation plot of the raw reflectance data in supplementary material. This plot is also shown below for convenience show that the variability of the reflected signal is markedly lower in the 550 nm region in both the case and control groups which explains the low p-value. Changes to the manuscript:  - Addition of Figure S3 in supplementary material. - Results: “The difference between cases and controls observed at wavelengths close to 550 nm (Fig. S2) approached statistical significance because the spectral variability within each group is lower in this wavelength range (Fig. S3).”

(8) The authors then performed a correction on the raw spectra by normalisation, using the average raw retinal spectrum from all subjects (PET + and PET -). This procedure takes out the large dynamic range and cancels effects from retinal features that are in common (eg. hemoglobin variations will average and cancel). The more subtle spectral features are now visualizable.	The reviewer makes an important remark regarding our inconsistent use of the term “normalisation”. What we were referring to in fact was centering the data about the average spectrum from all subjects. This is now clarified in the revised version of the manuscript (L 143, Fig. 2&3).
(9) Apparently the correction is a necessary step for seeing case-related differences of size well below the dynamic range of retinal reflectance spectra. In the main text the normalised spectra are said to highlight variations from retinal structures at the different locations, and also the large degree of inter-subject variability using uncorrected spectral data (also stated in Figure 2F-K caption). Actually, the SE (error bars) shown in the figure involve the variance of each case spectrum and the average spectrum, and would be determined using error propagation. These new errors would not be exclusive to original variability in the uncorrected spectra. (10) The variability shown on corrected spectra is large compared to changes in mean values about zero or between the test and control spectrum. Some features of the corrected spectra do in fact pertain to retinal structure variability, however the spectra also show a distinct feature related solely to case. Please note that Fig2F-K shows a consistent intensity inversion across wavelengths between	As we have now clarified the description regarding this “correction” (described above and done in response to the reviewer’s helpful suggestion), it is now apparent that centering the data does not affect calculation of the variability (SE). For instance, for any value “a” used for centering and any random sample X, the equation $SE(X) = SE(X-a)$ is always true. Accordingly, the error bars that appear in Figures 2F-K are correct. This is another astute observation. Although it could appear that there is a flip between cases and controls for sampling locations F1 and F2 in Fig. 2F-K, the difference is non-significant ($p > 0.4$) as per Fig. S2.

the test and control spectra, more so for locations F1 and F2. The curves are flipped. Normalisation using the full average from both groups (PET + and PET -) has produced opposing effects for each group. This is not mentioned by the authors. Such a trend in the corrected spectra would not come from variation in retinal structure so much as from a difference between test and control. If a spectral component from an AD biomarker were present in just one of the groups, normalization by an average taken over case and control would act, by cancellation, to leave a positive residual spectrum of the marker in one group, and a negative residual in the other. Spectra in Fig2F-K are consistent with this possibility.	
(11) This reviewer believes there is evidence for a biomarker in the authors' data and that it can be recovered without mitigation for effects of retinal structures on retinal spectra. Nonetheless, these more advanced analyses may be of interest to certain readers and are explained in a way these readers would appreciate.	We agree with the reviewer that there is a trend towards a significant difference between cases and controls using the uncorrected data, especially in the spectral region around 550 nm. The analyses performed in this study have reduced the variability in the data and show highly significant differences between cases and controls in two parts of the spectral data and overall when considered as a HS score.

REVIEWERS' COMMENTS:

Reviewer #1 (Remarks to the Author):

The authors addressed adequately the previous comments. No more concerns at this time.

Reviewer #2 (Remarks to the Author):

The authors have updated their manuscript with appropriate revisions suggested by this reviewer. By their spectral analysis methods, they showed there is a significant difference in the retinal reflectance over wavelength between PET + subjects with MCI and PET - control subjects, and a similarity in the spectral signatures of human PET + subjects and AB transgenic mice that is missing in control mice. These findings taken together would suggest that a spectral biomarker for the AB burden in Alzheimer's disease may be found in hyperspectral retinal images and may aid in predicting AB load. Direct evidence that the biomarker arises from AB is not easily discerned from this experiment (inconclusive match with in vitro scatter spectrum), so we are not sure if the biomarker represents a molecular marker or has a different origin such as from altered cell and tissue effects caused by the disease. This would be important for the authors to discuss, and may have been alluded to in line 401 of their revised paper. Please expand on the known influences on ocular reflectance, with molecular versus cell/tissue in mind. It seems this manuscript provides support for use of hyperspectral image data as a possible test for the presence of AB burden associated with Alzheimer's disease.

Response to Reviewers' comments:

Reviewer #1 (Remarks to the Author):

The authors addressed adequately the previous comments. No more concerns at this time.	Thank you
--	-----------

Reviewer #2 (Remarks to the Author):

The authors have updated their manuscript with appropriate revisions suggested by this reviewer. By their spectral analysis methods, they showed there is a significant difference in the retinal reflectance over wavelength between PET + subjects with MCI and PET - control subjects, and a similarity in the spectral signatures of human PET + subjects and AB transgenic mice that is missing in control mice. These findings taken together would suggest that a spectral biomarker for the AB burden in Alzheimer's disease may be found in hyperspectral retinal images and may aid in predicting AB load.	We appreciate this feedback with thanks.
Direct evidence that the biomarker arises from AB is not easily discerned from this experiment (inconclusive match with in vitro scatter spectrum), so we are not sure if the biomarker represents a molecular marker or has a different origin such as from altered cell and tissue effects caused by the disease. This would be important for the authors to discuss, and may have been alluded to in line 401 of their revised paper. Please expand on the known influences on ocular reflectance, with molecular versus cell/tissue in mind. It seems this manuscript provides support for use of hyperspectral image data as a possible test for the presence of AB burden associated with Alzheimer's disease.	We thank the reviewer for this feedback. A substantial part of the discussion relates to the point that we do not have conclusive evidence that the HS score is due to amyloid beta alone (extract (1) below). We have modified the wording of the abstract to indicate that other changes that occur in the retina in AD may be contributory (extract (2) below). We consider that this limitation has now been addressed adequately in the manuscript. (1) Discussion:...“We do not have conclusive evidence that the HS score is due to retinal Aβ alone. It is possible that the spectral effect observed in this study was due to other factors that are associated with brain PET Aβ status, such as iron accumulation, tau phosphorylation or inflammation. Any ocular constituent with a similar spectral profile to that measured for Aβ in solution could account for the observed effect (Supplementary Figure 12), provided that it was differentially distributed between Aβ PET+ cases and Aβ PET- controls. For instance, a recent large longitudinal study found that subtle changes in thickness of the retinal nerve fibre layer, measured using optical coherence tomography (OCT), were associated with cognitive decline.” (2) Abstract:...“ Studies of rodent models of Alzheimer's disease and of human tissues suggest that the retinal changes that occur in AD, including the accumulation of amyloid beta (Aβ), may serve as surrogate markers of brain Aβ levels. “